# Cucurbit[8]uril-based water-dispersible assemblies with enhanced optoacoustic performance for multispectral optoacoustic imaging

Yinglong Wu[1,4], Lihe Sun [2,4], Xiaokai Chen[1], Jiawei Liu[1], Juan Ouyang[2], Xiaodong Zhang [1], Yi Guo[1], Yun Chen[1], Wei Yuan[1], Dongdong Wang [1], Ting He[1], Fang Zeng[2], Hongzhong Chen [1,3] ✉, Shuizhu Wu [2] ✉ & Yanli Zhao [1] ✉

Organic small-molecule contrast agents have attracted considerable attention in the field of multispectral optoacoustic imaging, but their weak optoacoustic performance resulted from relatively low extinction coefficient and poor water solubility restrains their widespread applications. Herein, we address these limitations by constructing supramolecular assemblies based on cucurbit[8] uril (CB[8]). Two dixanthene-based chromophores (DXP and DXBTZ) are synthesized as the model guest compounds, and then included in CB[8] to prepare host-guest complexes. The obtained DXP-CB[8] and DXBTZ-CB[8] display red-shifted and increased absorption as well as decreased fluorescence, thereby leading to a substantial enhancement in optoacoustic performance. Biological application potential of DXBTZ-CB[8] is investigated after co-assembly with chondroitin sulfate A (CSA). Benefiting from the excellent optoacoustic property of DXBTZ-CB[8] and the CD44-targeting feature of CSA, the formulated DXBTZ-CB[8]/CSA can effectively detect and diagnose subcutaneous tumors, orthotopic bladder tumors, lymphatic metastasis of tumors and ischemia/reperfusion-induced acute kidney injury in mouse models with multispectral optoacoustic imaging.

Optoacoustic (OA, also called photoacoustic) imaging is a rising non-ionizing and non-invasive biomedical imaging modality whose principle relies on an alleged light-in and sound-out method[1]. An optoacoustic device transmits light energy to sample, receives ultrasound waves produced by the absorbers in the sample as the consequence of thermoelastic expansion, and then converts the collected acoustic data into optoacoustic images[2]. Compared with conventional optical or ultrasound imaging, optoacoustic imaging exhibits more significant advantages in imaging contrast, resolution and tissue penetration[3]. However, for some biological samples (e.g. blood, organs, and living bodies), optoacoustic signals generated by administrated contrast agents always overlap with those from the endogenous absorbers (e.g. hemoglobin and melanin), thus making it hard to precisely detect the target analyte with single-wavelength excitation. Multispectral optoacoustic imaging technique (MSOT) has been developed to address this challenge[4]. A multispectral optoacoustic imaging implement operates

[1]School of Chemistry, Chemical Engineering and Biotechnology, Nanyang Technological University, 21 Nanyang Link, Singapore 637371, Singapore. [2]Biomedical Division, State Key Laboratory of Luminescent Materials and Devices, Guangdong Provincial Key Laboratory of Luminescence from Molecular Aggregates, College of Materials Science and Engineering, South China University of Technology, Wushan Road 381, 510640 Guangzhou, China. [3]Institute of Pharmaceutics, School of Pharmaceutical Sciences (Shenzhen), Sun Yat-sen University, 518107 Shenzhen, China. [4]These authors contributed equally: Yinglong Wu, Lihe Sun. ✉e-mail: chenhzh58@mail.sysu.edu.cn; shzhwu@scut.edu.cn; zhaoyanli@ntu.edu.sg

by sequentially illuminating a sample with pulsed laser of multiple wavelengths and subsequently conducting spectral unmixing with mathematical algorithms, therefore the distribution as well as concentration of individual exogenous contrast agent or endogenous absorber in those biological samples can be imaged and quantified independently for accurate analysis of targets of interest[5]. Moreover, multispectral optoacoustic imaging system enables to provide more comprehensive information on the target of interest by rendering stacks of tomographic cross-sectional images as orthogonal projection images or direct volumetric imaging[6]. In the past few years, multispectral optoacoustic imaging has been developing expeditiously, eliciting a vast range of novel discoveries and applications[7–12].

Along with the rapid advancement of the imaging technique, tremendous endeavors have been devoted to developing versatile contrast agents including activatable ones for preclinical and clinical applications by using organic or inorganic chromophores[13–20]. The contrast agents with characteristic absorption spectra in the deep-red to near-infrared region, which displays significant differences from those of endogenous absorbers in tissue, are more beneficial for multispectral optoacoustic imaging, because their featured spectral profile with remarkable sharp peak(s) facilitates more accurate spectral unmixing, thus improving the sensitivity and specificity of imaging[21]. From this point of view, the contrast agents derived from organic small-molecule chromophores are favorable to definite identification in multispectral optoacoustic imaging, for that they typically display distinctive absorption peak(s) resulting from their well-defined chemical structure[22]. However, the relatively low extinction coefficient inherent in organic chromophores severely compromises their optoacoustic performance[21]. In addition, another challenge is that most of these organic chromophores are difficult to be dissolved in aqueous media owing to their rigid π-conjugated molecular backbones. To address this issue, chemical modification or physical encapsulation are the common methods to improve the water solubility/dispersibility of synthetic chromophores for preparation of optoacoustic contrast agents[23]. Unfortunately, the chemical modification and purification process is quite complicated and time-consuming, while physical encapsulation often leads to broadened as well as decreased absorption peak(s) and subsequent reduction in optoacoustic signal due to the random aggregation of contrast agents. Thence it is highly desirable to develop simple approaches to fabricating organic chromophores into highly absorbing contrast agents with improved optoacoustic performance and water dispersibility for biomedical applications.

In recent years, supramolecular self-assembled nanostructured materials have raised extensive attention in biomedical applications due to their ease of preparation via non-covalent interactions (e.g. host–guest inclusion, hydrogen bonding, electrostatic interaction, and π–π stacking)[24–31]. Particularly, macrocyclic supramolecular systems are more attractive because various properties (e.g. luminescence, water dispersibility, stability, biocompatibility, and bioavailability) of guest organic molecules can be ameliorated upon complexation with macrocyclic host molecules[32–35]. Among the various macrocyclic host compounds, cucurbit[8]uril (CB[8]) which possesses a rigid molecular structure, a large internal hydrophobic cavity, and a high negative charge density at the portals, can bind more strongly with guest molecules, especially positively charged ones, by means of the strong charge dipole, hydrogen bonding and hydrophilic/hydrophobic interactions[36–38]. Therefore, CB[8]-based supramolecular host–guest complexes have shown promising advantages in diverse fields including luminescent materials, cell imaging as well as cancer therapy[39–41]. However, macrocyclic CB[8] has yet been investigated to modulate optoacoustic performance of contrast agents by forming supramolecular host–guest complexes.

In this study, we report a strategy to simultaneously improve optoacoustic properties and water dispersibility of contrast agents by constructing cucurbit[8]uril-based supramolecular host–guest complexes, which can be utilized for multispectral optoacoustic imaging of CD44-overexpressing tumors and renal ischemia/reperfusion (I/R) injury after further being co-assembled with chondroitin sulfate A (CSA), as shown in Fig. 1. As a proof of concept, two positively charged chromophores, dixanthene derivatives (DXP and DXBTZ) were synthesized as the model guest compounds, which exhibited weak optoacoustic signals. After the host–guest interaction between the synthetic dixanthene derivatives and CB[8], the assembled water-dispersible linear supramolecular complexes exhibited enhanced optoacoustic signals owing to the improved light absorptivity and intensified non-radiative decay process of the excited state by suppressing the competitive fluorescence generation. To endow the dixanthene derivative-CB[8] complexes with targeting capability, they were further co-assembled with a negatively charged polysaccharide CSA, which has a high affinity for CD44 receptors[42–44]. Importantly, the formed ternary supramolecular nanoagents with spherical morphology maintained the enhanced optoacoustic signal. Moreover, our results demonstrate the formulated cucurbit[8]uril-based supramolecular nanoagents can efficiently accumulate in disease foci via the active CD44 targeting effect, and provide accurate detection and comprehensive visualization of tumors and renal I/R injury by multispectral optoacoustic imaging.

## Results

### Binding of macrocyclic CB[8]s with the synthetic chromophores as well as optical properties and morphologies of the resulting host–guest complexes

To construct CB[8]-based supramolecular assemblies as optoacoustic contrast agents, two model chromophore dixanthene derivatives (DXP and DXBTZ) were rationally designed as the guest molecules, by incorporating two cationic organic oniums (pyridinium and benzothiazolium) with different electron-withdrawing ability onto an electron-donating dixanthene core. These two model organic chromophores as well as their corresponding intermediates and control compounds were synthesized in accordance with the routes presented in Supplementary Fig. 1. These compounds were characterized by [1]H NMR, [13]C NMR as well as mass spectrometry, and the results are displayed in Supplementary Figs. 2–21.

Next, the host–guest molecular binding behaviors were investigated by monitoring the changes in absorption, optoacoustic performance, and fluorescence intensity of the synthetic dixanthene derivatives upon addition of CB[8] in aqueous solution (Fig. 2). These measurements were implemented after mixing the guest compound DXP (or DXBTZ) with varied amount of CB[8] in aqueous solution at 25 °C, as shown in Fig. 2a–d (or Fig. 2g–j). In the absence of CB[8], free DXP displayed an absorption peak at 608 nm (Fig. 2a) and intensive fluorescence emission at 760 nm (Fig. 2d), but weak optoacoustic signal at 680 nm (Fig. 2c). While upon stepwise addition of CB[8], the absorption peak of DXP red-shifted to 658 nm and its intensity steadily increased (Fig. 2a, b). Correspondingly, the optoacoustic signal at 680 nm gradually enhanced (Fig. 2c), but the fluorescent emission at 760 nm decreased remarkably (Fig. 2d). Compared with DXP, DXBTZ possessed a lower HOMO-LUMO energy gap resulting from the stronger electron-withdrawing capability of its binding moiety benzothiazolium (Supplementary Fig. 22), thus showing an absorption peak at longer wavelengths (Fig. 2g). Upon mixing with CB[8], DXBTZ also exhibited an enhanced absorption peak at 692 nm which is red-shifted from 662 nm, prominently increased optoacoustic signal at 692 nm as well as reduced fluorescence at 820 nm (Fig. 2g–j). Moreover, the photothermal conversion ability of the guest compound DXP (or DXBTZ) before and after complexation with CB[8] was estimated under 730 nm laser irradiation at 0.5 W cm⁻². It was found that the temperature of the mixed solution of DXP (or DXBTZ) and CB[8] elevated more than that of the solution of DXP (or DXBTZ) alone after continuous laser irradiation (Fig. 2e, k).

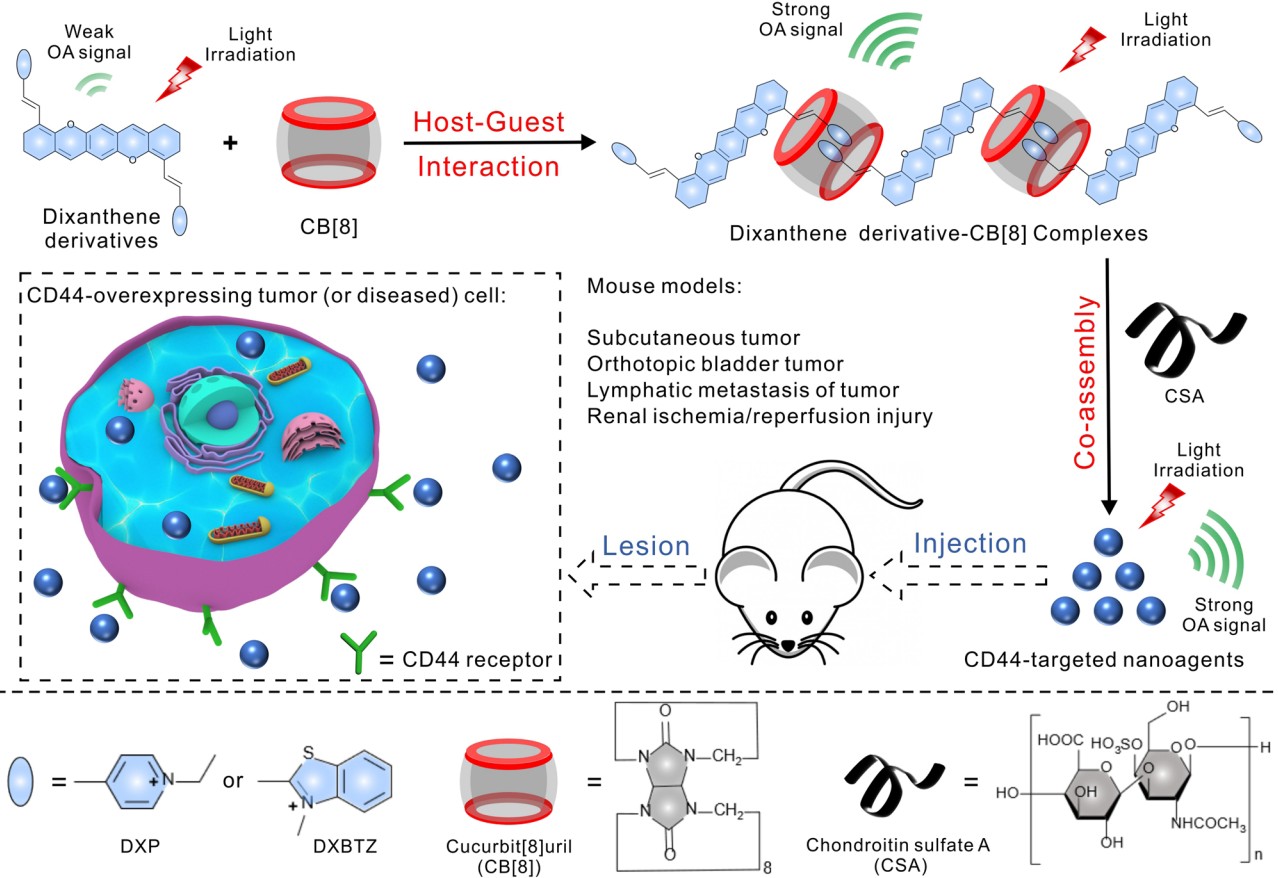

**Fig. 1 | Schematic illustration.** Construction of cucurbit[8]uril-based water-dispersible assemblies as contrast agents with enhanced optoacoustic performance for multispectral optoacoustic imaging of CD44-overexpressing tumors and renal I/R injury.

Optoacoustic effect relies on the efficiency of converting absorbed optical energy into thermal energy and then into detectable acoustic waves, therefore the optoacoustic performance of a chromophore is highly dependent on its absorption coefficient and deactivation pathways of its excited state[22]. As suggested by the results in Fig. 2, the improvement in optoacoustic performance of the chromophores after complexation with CB[8]s can be attributed to two aspects: on one hand, the increase in their absorption strengthens the uptake of optical energy; on the other hand, the reduction in their fluorescence contributes to decreasing the energy loss in the radiative decay of their excited state, which in turns intensifies the non-radiative transition for more heat generation and the subsequent enhancement in optoacoustic signal. It is worth noting that, when the equimolar amount of CB[8] was added, the optical properties of dixanthene derivatives leveled off, indicating all the guest compounds were included in CB[8] and formed the host–guest complexes. In addition, the wavelength-dependent OA signal of DXBTZ-CB[8] exhibited a maximum optoacoustic peak at around 692 nm (Supplementary Fig. 23), which is consistent with its maximum absorption wavelength (Fig. 2g). However, DXP-CB[8] showed the maximum OA signal intensity at around 680 nm as our optoacoustic tomography system can only detect optoacoustic signal in the range of 680-980 nm. Moreover, the extinction coefficients for the resulting host–guest complexes DXP-CB[8] and DXBTZ-CB[8] were calculated to be $6.9 \times 10^4 \, M^{-1} cm^{-1}$ at 680 nm and $5.8 \times 10^4 \, M^{-1} cm^{-1}$ at 692 nm respectively, which were much higher than those for DXP ($3.5 \times 10^4 \, M^{-1} cm^{-1}$ at 680 nm) and DXBTZ ($3.6 \times 10^4 \, M^{-1} cm^{-1}$ at 692 nm) alone.

Additionally, the morphology and size distribution of these binary host–guest complexes were determined by transmission electron microscope (TEM) and dynamic light scattering (DLS) experiments.

Both of the binary host–guest complexes DXP-CB[8] and DXBTZ-CB[8] presented the nanorod-like morphology in the TEM images shown in Fig. 2f, l. The average hydrodynamic diameters of DXP-CB[8] and DXBTZ-CB[8] were determined as 730 nm and 510 nm respectively. Moreover, these two binary complexes are stable for several weeks at room temperature (Supplementary Fig. 24). These results clearly suggest that these two model organic chromophores (DXP and DXBTZ) can serve as the guest compounds to construct stable supramolecular host–guest complexes with enhanced optoacoustic signals in aqueous solution, which have the potential to be employed for multispectral optoacoustic imaging in biomedical fields.

## Mechanism of complexation between CB[8] and dixanthene derivatives

To elucidate the effect of complexation with CB[8] on the properties of dixanthene derivatives, the binding mode of the resulting host–guest complexes was characterized. As can be seen from the Job's plots in Supplementary Fig. 25, dixanthene derivative-CB[8] complexes ostensibly adopted a stoichiometric ratio of 1:1. By using a nonlinear least-squares curve-fitting method, the association constants (Ks) for complexation of CB[8] with DXP or DXBTZ were determined as $4.36 \times 10^8 \, M^{-1}$ and $5.84 \times 10^8 \, M^{-1}$ respectively, which would ensure high stability of the obtained dixanthene derivative-CB[8] complexes in aqueous environment for further functionalization and applications (Supplementary Fig. 26).

It has been reported that CB[8] possesses a portal diameter of 0.69 nm as well as a cavity dimeter of 0.88 nm, and can include one or two guest molecules with positive charge, depending on the size of the binding moiety in guest compounds[45]. Since one dixanthene derivative molecule has two cationic binding moieties, there might be three

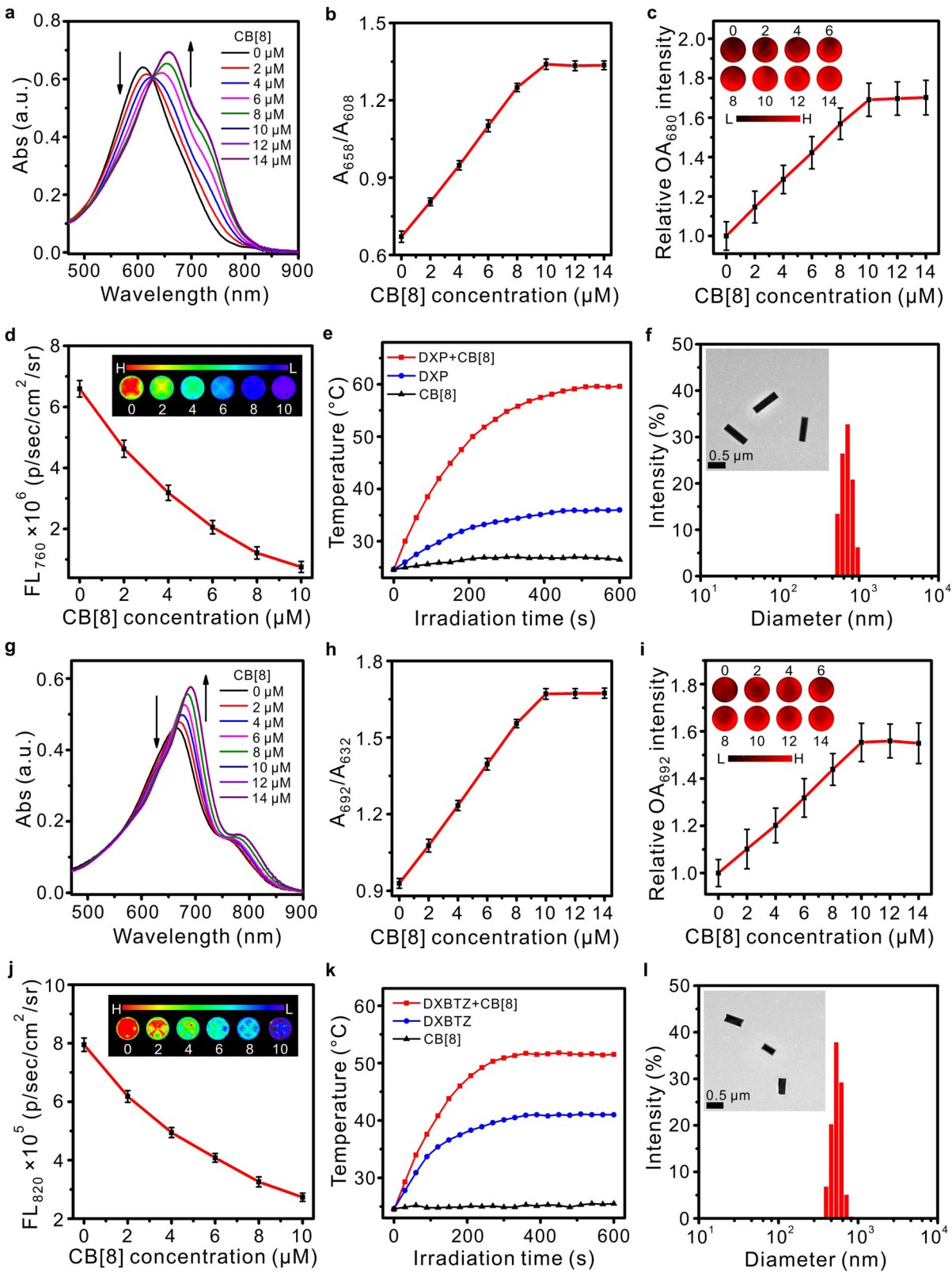

binding modes of dixanthene derivative-CB[8] complexes: simple 2:2 binding mode, stacked n:n binding mode and sled n:n binding mode (Fig. 3a). Simple 2:2 binding mode as well as stacked n:n binding mode can easily lead to H-aggregation of dixanthene derivative molecules, causing a blue shift in absorption. On the contrary, sled binding mode typically induces dixanthene derivative molecules to form J-aggregate complexes with red-shifted absorption. Considering the results in

Fig. 2, inclusion of DXP or DXBTZ into CB[8] prompted a red shift in absorption, so these binary complexes should adopt the sled n:n binding mode. Inclusion of dixanthene derivatives into CB[8] following the sled binding mode would result in linear host–guest assembly units, and then many assembly units could further stack with each other for the final nanorod-like morphology (Fig. 2f, l). Moreover, geometries of the guest compounds DXP and DXBTZ were also

**Fig. 2 | Effect of complexation with the macrocyclic host compound CB[8] on properties and morphologies of the guest compounds DXP and DXBTZ in aqueous solution. a, g** Absorption spectra for the guest compounds DXP and DXBTZ (10 µM) after mixing with different concentrations of CB[8] (0–14 µM). **b, h** Curves for absorbance ratio as a function of CB[8] concentration ($n = 3$ independent experiments). **c, i** Changes in relative optoacoustic intensity of DXP and DXBTZ (10 µM) after mixing with different concentrations of CB[8] (0–14 µM; $n = 3$ independent experiments). The insets present typical OA images of DXP and DXBTZ after mixing with different amounts of CB[8] in phantom (680 nm and 692 nm were selected as the excitation wavelength for **c** and **i**, respectively).

**d, j** Relationship between the fluorescence intensity and the concentration of added CB[8]. The insets display typical fluorescence images of DXP and DXBTZ (10 µM) with varied CB[8] concentrations (0–10 µM) in a six-well plate (640 nm and 675 nm were selected as the excitation wavelength for **d** and **j**, respectively; $n = 3$ independent experiments). **e, k** Photothermal conversion behavior of DXP and DXBTZ (10 µM) before and after complexation with CB[8] (10 µM) under 730 nm irradiation at 0.5 W cm$^{-2}$. **f, l** Hydrodynamic diameter distribution and representative transmission electron microscopic images (insets) for a sample of dixanthene derivatives (10 µM) mixed with CB[8] (10 µM). Data are presented as mean values ±standard deviation (SD). Source data are provided as a Source Data file.

optimized by density functional theory (DFT) calculations to verify their complexation, and the atomic coordinates were provided in Supplementary Tables 1 and 2. The maximum lateral dimensions of the binding moiety pyridinium and benzothiazolium in dixanthene derivatives were calculated to be 0.43 nm and 0.50 nm respectively, whereas the maximum lateral dimensions of the dixanthene core was determined as 1.42 nm (Fig. 3b, c). Therefore, CB[8] can only include the binding moieties (pyridinium and benzothiazolium) rather than the dixanthene core in the guest compounds. Moreover, the steric hindrance of the two saturated methylene groups prevents H-aggregation of dixanthene derivatives upon complexation with CB[8]. The simulation results also support the sled binding mode of dixanthene derivative-CB[8] complexes (Supplementary Fig. 27).

To verify the proposed mechanism, several NMR experiments were carried out. In the $^1$H NMR spectra of dixanthene derivatives with varied amounts of cucurbit[8]uril, clear and regular shifts of the proton peaks were observed, confirming the complexation between cucurbit[8]urils and dixanthene derivatives (Supplementary Fig. 28 and 29). For DXP and DXP-CB[8], the assignment of proton peaks was analyzed based on DQF-COSY and HSQC (Supplementary Figs. 30 and 31). As shown in Supplementary Fig. 32, the peaks at $\delta = 8.4$ and 7.5 are assigned to the protons of pyridine rings (1 and 3); the peaks at $\delta = 7.5$ and 6.1 are assigned to the protons of alkene moiety (2 and 6); and the peaks at $\delta = 4.3$ and 1.5 are assigned to the protons of aliphatic moiety (7 and 11). In general, the nuclear Overhauser effect (NOE) occurs when the distance between two protons is <5 Å[46]. Thus, 2D NOESY is often used to study the supramolecular stereo information. Within the DXP molecule, most of the distance between the aliphatic chain proton (7 or 11) and the pyridine ring proton (1 or 3) is already <5 Å (Supplementary Fig. 33). In contrast, the distance between the aliphatic chain proton (7 or 11) and the alkene moiety proton (2) is much >5 Å in a single DXP molecule, so that the appearance of NOE correlation peak between these protons could theoretically verify the interaction of DXP molecules in the CB[8] cavity. However, we found the NMR signal of Proton 2 of DXP shifted upfield and overlapped with those of proton 1 and 3 when DXP complexed with CB[8]. For this reason, the NOESY spectrum of DXP-CB[8] did not show intuitive evidence for the interaction mode of DXP molecules in the CB[8] cavity (Supplementary Fig. 34). Nevertheless, the $^1$H NMR spectra of DXP and DXP-CB[8] provided insights into the proposed binding mode. As shown in Supplementary Fig. 32, all peaks of the pyridine ring protons (1 and 3) and the aliphatic moiety protons (7 and 11) underwent a pronounced upfield shift after DXP complexed with CB[8], which caused by the shielding effect of another DXP molecule in the cavity, and therefore indicating the formation of J-aggregates[47,48]. The shielding effect of the CB[8] cavity on the other side of the DXP molecule also contributes to the upfield shift, illustrating the interaction between CB[8] and DXP. This host–guest interaction was also confirmed by DOSY experiments, and diffusion coefficient of DXP-CB[8] was determined as $1.7115 \times 10^{-10}$ m$^2$ s$^{-1}$ (Supplementary Fig. 35).

Similarly, the peak assignment of DXBTZ and DXBTZ-CB[8] was also analyzed based on DQF-COSY and HSQC (Supplementary Figs. 36 and 37), and the results were shown in Supplementary Fig. 38. Although the peaks assigned to aromatic protons of DXBTZ only

exhibited broadened signs due to π–π stacking of their dixanthene core in the solvent, the peak belonging to the hydrophilic methyl group proton (11) remained very clear. As seen in Supplementary Figs. 39 and 40, the distance between the aromatic proton 4 and methyl group proton 11 was >5 Å in a single DXBTZ molecule, while the cross peak between these two protons appeared in the NOESY spectrum of DXBTZ-CB[8], demonstrating that DXBTZ molecules could only form J-aggregates to reduce the spacing between proton 4 of one DXBTZ molecule and proton 11 of the other one in the CB[8] cavity. Moreover, the $^1$H NMR changes of DXBTZ before and after complexion with CB[8] could also provide evidence for the proposed sled n:n binding mode. As seen in Supplementary Fig. 37, the broad NMR peaks of DXBTZ transformed into sharp and clear ones upon the addition of CB[8], suggesting the complexation between DXBTZ and CB[8]. At the same time, the peak assigned to proton 11 underwent an upfield shift, indicating the methyl group was strongly shielded by the host CB[8] molecule and the benzothiazole ring of the other DXBTZ molecule in the CB[8] cavity. In addition, the diffusion coefficient of DXBTZ-CB[8] was determined as $1.5115 \times 10^{-10}$ m$^2$ s$^{-1}$ according to the 2D DOSY spectrum (Supplementary Fig. 41). Therefore, it can be deduced that the guest molecule DXBTZ adopts a sled n:n interaction mode when complexing with the host molecule CB[8].

Additionally, only CB[8] could elicit significant bathochromic shifts in absorption of dixanthene derivatives and enhancement in optoacoustic signal (Fig. 3d, e, g, h), because CB[7] or CB[6] could not include two guest molecules to form complexes with extended conjugation in their small cavity. Control guest compounds (XP and XBTZ) with single cationic binding moiety were also synthesized and employed for complexation with CB[8]. However, no apparent redshifted absorption peak could be observed after mixing CB[8] with the control guest compound (Fig. 3f, i). These results further confirm the mechanism of complexation between CB[8] and dixanthene derivatives with two cationic binding moieties.

## Preparation of CB[8]-based optoacoustic nanoagents with CD44 targeting ability

Before application of the dixanthene derivative-CB[8] complexes in biomedical imaging, the positive charges on their surface and their size distribution should be further adjusted to avoid rapid clearance from the blood system[49]. Compared with DXP-CB[8], DXBTZ-CB[8] exhibited a characteristic absorption peak and the corresponding maximum optoacoustic signal at 692 nm within the detection range (680-980 nm) of our multispectral optoacoustic imaging system, which facilitated unambiguous identification in the spectral unmixing process, and thus were selected for further functionalization and application. Herein, a natural polysaccharide, chondroitin sulfate A with negative charges was chosen for co-assembly with DXBTZ-CB[8] complex, owing to its good biodegradability, high biocompatibility, low immunogenicity, and CD44 targeting ability[42–44]. As can be seen in Fig. 4a, the zeta potential of DXBTZ-CB[8] was detected as +16.7 mV in the absence of CSA, and then gradually changed from positive to negative with increasing the concentration of added CSA. When the CSA concentration exceeded 8 µg mL$^{-1}$, both the zeta potential and the average diameter of the resulting mixture leveled off, indicating that a

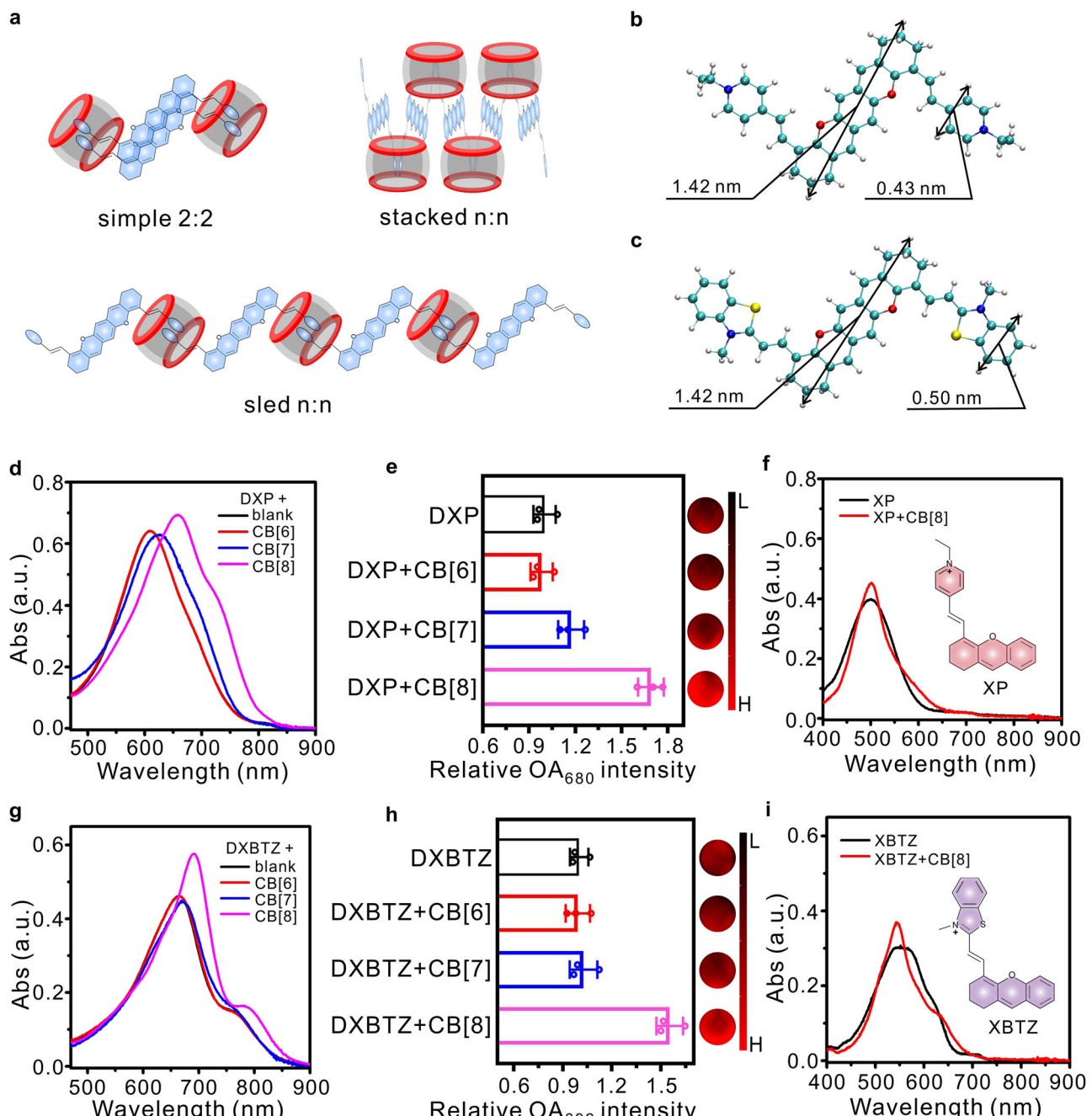

**Fig. 3 | Verification for the dixanthene derivative-CB[8] complexation mechanism. a** Illustration for three feasible binding modes between dixanthene derivatives and cucurbit[8]uril. **b, c** Structure of DXP and DXBTZ as optimized by density functional theory calculations. The maximum lateral dimensions of the entire molecule (DXP and DXBTZ) and their corresponding binding moieties were calculated, respectively. **d, g** Absorption spectra for the model guest compound dixanthene derivatives (10 μM) upon mixing with equivalent concentration of macrocyclic CB[6], CB[7], or CB[8]. **e, h** Left side: relative OA intensity of DXP and DXBTZ (10 μM) upon mixing with equivalent concentration of CB[6], CB[7], or CB[8] (*n* = 3 independent experiments). Right side: the typical OA images of dixanthene derivatives with equivalent concentration of CB[6], CB[7], or CB[8] in phantom (680 nm and 692 nm were selected as the excitation wavelength for **e** and **h**, respectively). **f, i** Absorption spectra for the control guest compound xanthene derivatives (10 μM) upon mixing with CB[8] (5 μM). Data are presented as mean values ± SD. Source data are provided as a Source Data file.

new stable nanostructure had been co-assembled from DXBTZ-CB[8] and CSA. The co-assembly process of DXBTZ-CB[8] and CSA was monitored by TEM and DLS experiments, and the results were shown in Supplementary Fig. 42. Driven by the negative charges of CSA, the nanorods stacked by positively charged binary unit DXBTZ-CB[8] began to dissociate into small fragments, which then co-assembled with CSA into slack nanostructures via electrostatic attraction. Upon the prolongation of incubation time, the enhanced electrostatic

interaction and the formed hydrogen bonds between DXBTZ-CB[8] and CSA enabled the co-assembled nanoparticles to gradually change from loose to dense, and finally present a spherical shape with an average diameter of 186 nm (Fig. 4b). Furthermore, the negative zeta-potential of obtained DXBTZ-CB[8]/CSA nanoagents (Fig. 4a) suggests that CSA is distributed on the surface for its potential CD44 targeting capability, which is similar to the surface engineering of host–guest complexes[39]. Importantly, the ternary assembly DXBTZ-CB[8]/CSA

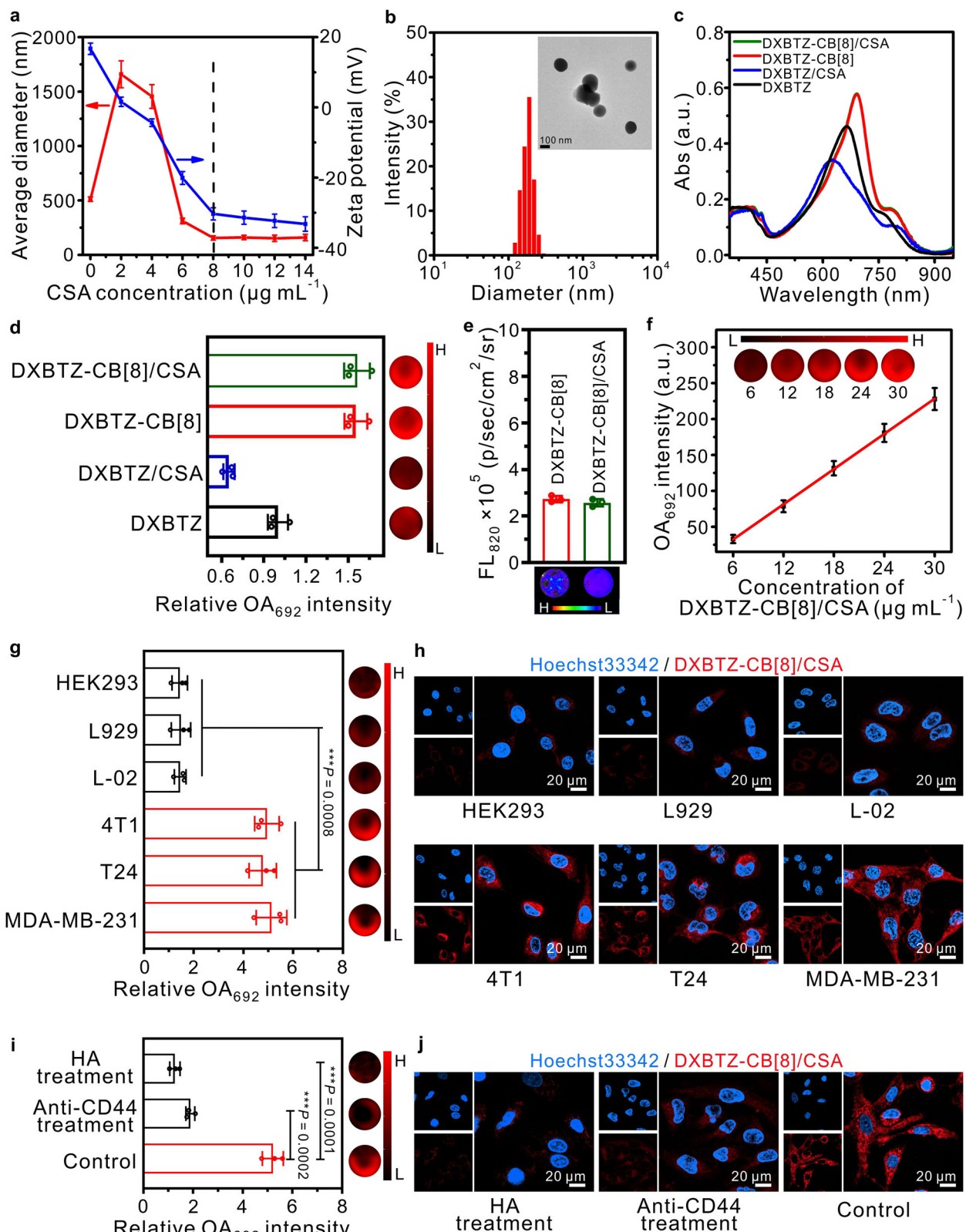

exhibited similar absorption and optoacoustic performance to the binary complex DXBTZ-CB[8], while directly mixing DXBTZ with CSA only resulted in the products (DXBTZ/CSA) with blue-shifted absorption and decreased optoacoustic signals due to random $\pi-\pi$ stacking of chromophores (Fig. 4c, d). Moreover, the co-assembly between DXBTZ-CB[8] and CSA also had little impact on the fluorescence emission intensity, as can be seen in Fig. 4e. These results demonstrate

that the preparation of DXBTZ-CB[8]/CSA nanoagents via co-assembly of CSA with DXBTZ-CB[8] complexes only break the stacking of the binary assembly units, but not destroy the inclusion between DXBTZ and CB[8]. Additionally, the optoacoustic intensity of DXBTZ-CB[8]/CSA at 692 nm was linearly related to its concentration, as shown in Fig. 4f. Although other macromolecular substances (e.g. FBS) may slowly dissociate the supramolecular nanoagents, DXBTZ-CB[8]/CSA

**Fig. 4 | Properties of the CD44-targeted supramolecular optoacoustic nanoagent DXBTZ-CB[8]/CSA. a** Average diameters and zeta potentials of the nanoagents obtained by mixing the DXBTZ-CB[8] (10 μM) with varied amounts of CSA (0–14 μg mL⁻¹) in aqueous solution for 30 min (*n* = 3 independent experiments). **b** Hydrodynamic diameter distribution and a representative transmission electron microscopic image (inset) for the sample of DXBTZ-CB[8] (10 μM) 30 min after addition of CSA (8 μg mL⁻¹). **c** Absorption spectra for DXBTZ, DXBTZ-CB[8], DXBTZ-CB[8]/CSA as well as the mixture (DXBTZ/CSA) of DXBTZ and CSA. **d** Left side: relative OA intensity at 692 nm of DXBTZ, DXBTZ-CB[8], DXBTZ-CB[8]/CSA as well as the mixture (DXBTZ/CSA) of DXBTZ and CSA in phantom (*n* = 3 independent experiments). Right side: representative optoacoustic images of the corresponding groups ([DXBTZ] = [CB[8]] = 10 μM, [CSA] = 8 μg mL⁻¹). **e** Fluorescence intensity (top row) of DXBTZ-CB[8] as well as DXBTZ-CB[8]/CSA (*n* = 3 independent experiments) and their corresponding fluorescence images (bottom row). **f** Optoacoustic intensity of DXBTZ-CB[8]/CSA at 692 nm in phantom as a function of mass concentration (*n* = 3 independent experiments). The inset presents typical OA images of DXBTZ-CB[8]/CSA at different concentrations. **g** Typical optoacoustic images (right side) for HEK293, L929, L-02, 4T1, T24, and MDA-MB–231 cells incubated with DXBTZ-CB[8]/CSA (50 μg mL⁻¹) in phantom as well as their corresponding relative OA intensities at 692 nm (*n* = 3 independent experiments) (left side). **h** Typical confocal fluorescence images for HEK293, L929, L-02, 4T1, T24, and MDA-MB-231 cells incubated with DXBTZ-CB[8]/CSA (50 μg mL⁻¹) and Hoechst33342. **i** Typical optoacoustic images (right side) for MDA-MB-231 cells or CD44 inhibitor HA or anti-CD44 antibody pretreated cells incubated with DXBTZ-CB[8]/CSA (50 μg mL⁻¹) in phantom, as well as their corresponding relative OA intensities at 692 nm (*n* = 3 independent experiments; left side). **j** Typical confocal fluorescence images for MDA-MB-231 cells or CD44 inhibitor HA or anti-CD44 antibody pretreated cells incubated with DXBTZ-CB[8]/CSA (50 μg mL⁻¹) and Hoechst33342. Data are presented as mean values ± SD. Statistical significance was determined by two-tailed *t* test. \*\*\**P* < 0.001. Source data are provided as a Source Data file.

also manifested considerable stability within 3 days, which is quite promising for targeted imaging and subsequent metabolic clearance in biological applications (Supplementary Fig. 43).

## Validation of the CD44-targeting ability of DXBTZ-CB[8]/CSA by cell imaging

Prior to validating the CD44-targeting ability of DXBTZ-CB[8]/CSA by cell imaging, its cytotoxicity was assessed using MTT assays. As can be seen in Supplementary Fig. 44, the nanoagent DXBTZ-CB[8]/CSA displayed low cytotoxicity even at the concentration of 50 μg mL⁻¹, implying that DXBTZ-CB[8]/CSA is well suited to cell imaging. Next, the time-dependent uptake of DXBTZ-CB[8]/CSA by MDA-MB-231 cells was evaluated by flow cytometry and changes in optical density at 692 nm (OD₆₉₂). Approximately 60 min after incubation, the fluorescence intensity and OD₆₉₂ in cells leveled off, indicating cellular uptake of DXBTZ-CB[8]/CSA reached the maximum level (Supplementary Figure 45). Afterwards, the utility of the nanoagent DXBTZ-CB[8]/CSA for optoacoustic and fluorescence imaging in living cells was investigated, and the mean OA intensities of different cells in phantom were also recorded. As seen from Fig. 4g, h, upon incubation with DXBTZ-CB[8]/CSA, obvious OA signals and fluorescence could be observed in 4T1, T24, and MDA-MB-231 cells, which has been reported to overexpress CD44 receptors[50,51], whereas there were quite weak OA signals and fluorescence in the normal HEK293, L929, and L-02 cells. It suggests the large amounts of CD44 receptors on the surface of MDA-MB-231 cells enable higher uptake of DXBTZ-CB[8]/CSA via efficient receptor-mediated endocytosis. Furthermore, CD44 inhibition studies were carried out to validate the CD44-mediated cellular internalization of DXBTZ-CB[8]/CSA. As can be seen in Fig. 4i, j, the MDA-MB-231 cells pretreated with excess free hyaluronic acid (HA, a competitive inhibitor) or anti-CD44 antibody exhibited markedly lower optoacoustic signal and fluorescence than the untreated ones, which confirmed the endocytosis pathway of DXBTZ-CB[8]/CSA was mainly through CD44 receptor-mediated internalization. These observations signify DXBTZ-CB[8]/CSA possesses an excellent targeting capability to image and identify CD44-overexpressed abnormal cells.

## Multispectral optoacoustic imaging of subcutaneous tumor in a mouse model using the nanoagent DXBTZ-CB[8]/CSA

Inspired by the above results, the biosafety of DXBTZ-CB[8]/CSA was evaluated before multispectral optoacoustic imaging in mice. From Supplementary Fig. 46, no notable differences in the body weights could be observed between the control group administered with saline and the group intravenously injected with the nanoagent DXBTZ-CB[8]/CSA. Moreover, there were no apparent histopathological morphology abnormalities among the organs from the control group as well as the group administered with DXBTZ-CB[8]/CSA (Supplementary Fig. 47). Furthermore, blood samples were collected from the mice 7 days after injection of saline or DXBTZ-CB[8]/CSA, and representative blood routine indicators as well as serum biomedical parameters reflecting liver or kidney functions were determined. As can be seen in Supplementary Fig. 48, the indexes were similar among the control group and the group administered with DXBTZ-CB[8]/CSA. The results of body weight measurement, organ histology examination, blood routine tests, and serum biochemistry assays denote that DXBTZ-CB[8]/CSA is of good biosafety. In addition, the optoacoustic properties of DXBTZ-CB[8]/CSA were compared with a commercially available contrast agent, PEG₅₀₀₀-functionalized gold nanorods (AuNR-PEG) at the same mass concentration. As shown in Supplementary Fig. 49, although these two contrast agents have similar absorption and optoacoustic spectra in the 680–900 nm wavelength range, DXBTZ-CB[8]/CSA displayed superior photostability and generated higher optoacoustic signal output than AuNR-PEG both in vitro and in vivo. The results imply that the nanoagent DXBTZ-CB[8]/CSA can exhibit higher signal-to-noise ratio in multispectral optoacoustic imaging and is more suitable for long-term tracking and monitoring.

To explore the capability of the CD44-targeted nanoagent DXBTZ-CB[8]/CSA to detect tumors by means of multispectral optoacoustic imaging, we first established a mouse model of subcutaneous tumor by injecting MDA-MB-231 cells onto the back of mice. Then the tumor-bearing mice were intravenously injected with DXBTZ-CB[8]/CSA, and subjected to multispectral optoacoustic imaging by a commercial MSOT system (Fig. 5a). Eight wavelengths corresponding to the major inflection points in the absorption curves of DXBTZ-CB[8]/CSA, oxyhemoglobin, and deoxyhemoglobin, were selected to excite the DXBTZ-CB[8]/CSA and endogenous hemoglobin for acquisition of the total optoacoustic signals, followed by spectral unmixing to distinguish the contribution of DXBTZ-CB[8]/CSA, which enabled generation of MSOT images for the tumor region by overlaying the optoacoustic signal (rainbow scale) of the supramolecular nanoagent DXBTZ-CB[8]/CSA on single-wavelength anatomical images (gray scale). MSOT images in Fig. 5b displayed the time-resolved metabolism behavior of DXBTZ-CB[8]/CSA in tumor. The optoacoustic signal of DXBTZ-CB[8]/CSA at the tumor site increased gradually after injection and reached a maximum at 12 h (Fig. 5b). Moreover, the orthogonal MIP image was generated through rendering the obtained cross-sectional MSOT images, which enabled to precisely pinpoint the tumor site, as shown in Fig. 5c. Afterwards, the OA signal gradually decreased over time, as can be observed from Fig. 5d. These results implied that DXBTZ-CB[8]/CSA primarily reached and accumulated in the tumor via active CD44-targeting effect within 12 h post intravenous injection, while underwent obvious metabolic clearance from the tumor after 12 h. As seen in Supplementary Fig. 50, ~90% of the injected nanoagents was excreted from the body through metabolism into feces and urine within 7 days of administration. In addition, some tumor-bearing

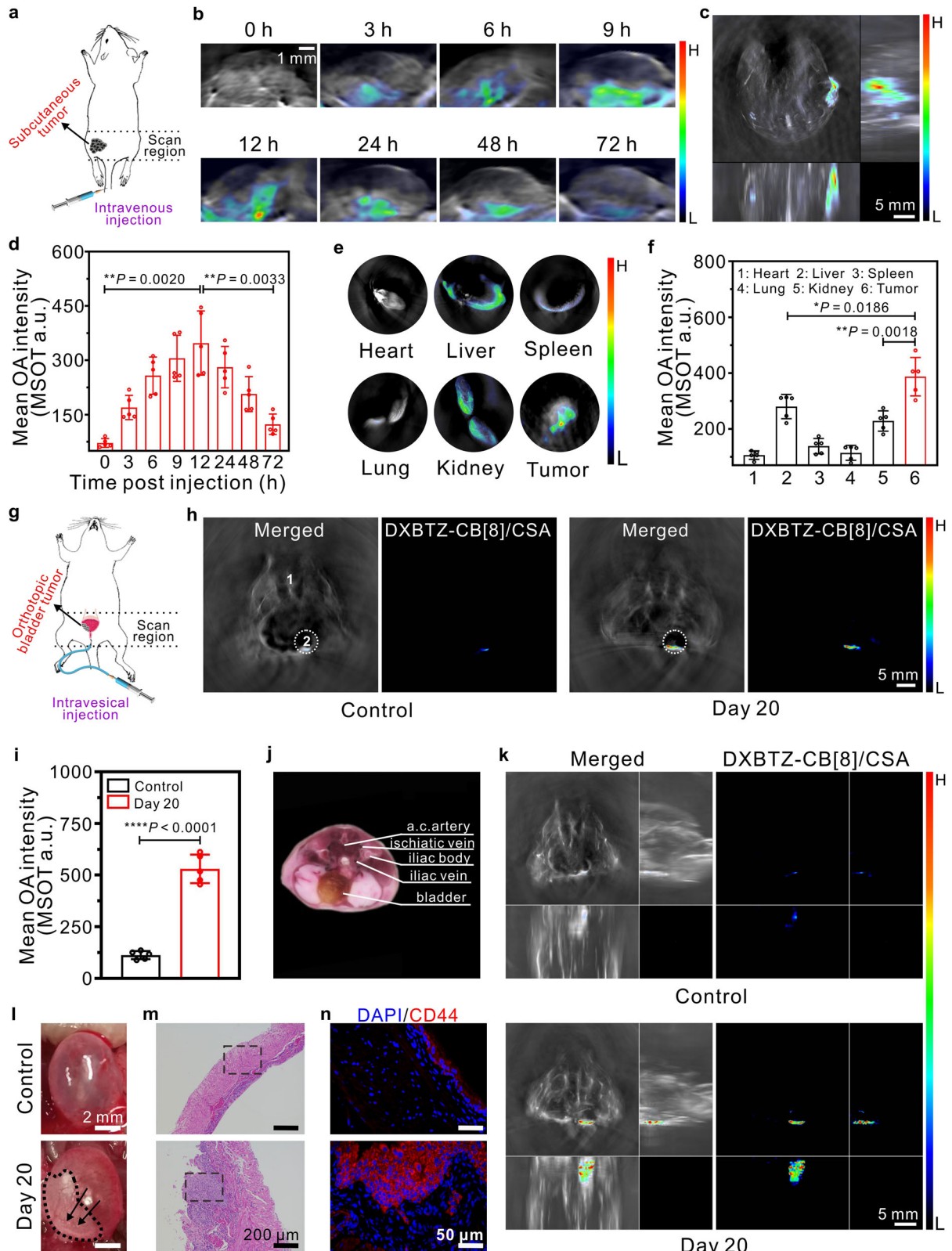

mice were euthanized 12 h after intravenous administration of DXBTZ-CB[8]/CSA, and their major organs as well as tumors were dissected for ex vivo imaging. As revealed by Fig. 5e, f, the tumors exhibited much more evident optoacoustic signals than other organs, thus proving the capability of the nanoagent DXBTZ-CB[8]/CSA to target CD44-overexpressing tumors. The observations demonstrate that DXBTZ-CB[8]/CSA is capable of targeting cancer cells with overexpressed

CD44 receptors in living mice, and detecting as well as positioning the tumors through multispectral optoacoustic imaging.

**Multispectral optoacoustic imaging of orthotopic bladder tumor in a mouse model using the nanoagent DXBTZ-CB[8]/CSA**
Bladder cancer is one of the most concerning urinary tract malignancies worldwide, giving a high mortality rate[52]. Hence, developing an

**Fig. 5 | Application of DXBTZ-CB[8]/CSA in the mouse model of subcutaneous tumor and orthotopic bladder tumor. a** Schematic illustration for multispectral optoacoustic imaging of subcutaneous tumor in mice after intravenous injection of DXBTZ-CB[8]/CSA. **b** Typical MSOT images for the tumor region of a mouse in the supine position at different time points after intravenous injection of DXBTZ-CB[8]/CSA (46.7 mg kg⁻¹). **c** A representative z-stack orthogonal MIP MSOT image for a mouse at 12 h after intravenous injection of DXBTZ-CB[8]/CSA. **d** Mean MSOT intensities of the tumor region at varied time points after intravenous injection of DXBTZ-CB[8]/CSA ($n = 5$ animals per group). **e** Representative MSOT images for the tumor and major organs excised from a mouse 12 h after intravenous injection of DXBTZ-CB[8]/CSA. **f** Mean optoacoustic intensities of the tumor and major organs of mice at 12 h upon intravenous injection of DXBTZ-CB[8]/CSA ($n = 5$ animals per group). **g** Schematic illustration for multispectral optoacoustic imaging of orthotopic bladder tumor in mice after intravesical injection of DXBTZ-CB[8]/CSA via a catheter. **h** Typical cross-sectional MSOT images for the control group (mice 20 days post intravesical injection of saline) and the orthotopic bladder tumor model group (mice 20 days post intravesical inoculation of T24 cells). The mice were placed in the prone position for imaging. Left panel: overlay of DXBTZ-CB[8]/CSA optoacoustic signal onto the grayscale image with anatomical information.

Right panel: spectrally unmixed optoacoustic signal of DXBTZ-CB[8]/CSA. Organ tags: 1: artery; 2: bladder. The mouse bladder region was delimited with a white dotted circle (ROI). **i** Mean MSOT intensities at ROI for the control mice and the mice with orthotopic bladder tumor at 1 h upon intravesical injection of DXBTZ-CB[8]/CSA (15.6 mg kg⁻¹; $n = 5$ animals per group). **j** Cryosection image of a female mouse corresponding to the cross-section location in **h**. **k** Orthogonal MIP MSOT images for the control mouse and the mouse 20 days post intravesical inoculation of T24 bladder cancer cells. Scale bar: 5 mm. **l** Photos for bladders from the control mouse and the mouse 20 days post intravesical inoculation of T24 bladder cancer cells. Tiny solid nodules (pointed by black arrows) and lesions (shown in a black dashed circle) could be clearly observed on the bladder wall of tumor-bearing mice. **m** Typical H&E-stained section for bladders from the control group and the orthotopic bladder tumor model group. Scale bar: 200 μm. The magnified images for the areas in the black dotted box are shown in Supplementary Fig. 51. **n** Immunofluorescence images of anti-CD44-stained bladder sections from the control group and the orthotopic bladder tumor model group. Scale bar: 50 μm. Data are presented as mean values ± SD. Statistical significance was determined by two-tailed $t$ test. *$P < 0.05$, **$P < 0.01$, and ****$P < 0.0001$. Source data are provided as a Source Data file.

effective approach to accurately detect and diagnose bladder cancer is of great importance. CD44 receptor has been found to be much more highly expressed in bladder tumors than that in normal tissues, making it an exceptional target for bladder cancer diagnosis[12,53]. Herein, we created a mouse model of orthotopic bladder tumor by intravesical injection of T24 cells (human bladder carcinoma cell line) into the mice immediately after chemical injury to the bladder urothelium[54], and then employed the nanoagent DXBTZ-CB[8]/CSA to diagnose orthotopic bladder tumor via targeting the overexpressed CD44 receptors of bladder cancer cells (Fig. 5g). The results pertinent to MSOT imaging of the orthotopic bladder tumor are shown in Fig. 5h–n. In addition, a cryosection image corresponding to the cross-section of a female mouse bladder is presented in Fig. 5j. As seen from Fig. 5h, i, there is a prominent MSOT signal from DXBTZ-CB[8]/CSA in the cross-section of the mouse 20 days post bladder cancer cell inoculation. Whereas for the healthy control mouse, only a weak signal can be observed. By referring to the cryosection image (Fig. 5j), it is evident that DXBTZ-CB[8]/CSA signal is at the bladder region. The orthogonal MIP images for the mice 20 days following intravesical injection of T24 cancer cells could be further obtained through rendering a series of acquired cross-sectional MSOT images. As shown in Fig. 5k, the location of the bladder tumor can be distinctly visualized by the generated orthogonal MIP image. The above results suggest that the nanoagent DXBTZ-CB[8]/CSA can be used to diagnose the orthotopic bladder tumor.

After the healthy (control) and tumor-bearing mice were killed and dissected, their bladders were collected and photographed. As shown in Fig. 5l, the bladder wall of the healthy mouse is transparent without any lesions. Whereas for the mice with orthotopic tumor, the bladder has apparent lesions as well as tiny solid nodules (tumors), and the bladder wall became less stretchy and elastic owing to the tumor formation within the bladder. Importantly, the location of the tumors matched well with that in the orthogonal MIP MSOT images. Moreover, blood in the urine of tumor-bearing mice could be clearly visualized with naked eyes, which is called hematuria, a common symptom of bladder cancer. In addition, H&E staining for the bladder tissue sections from the control and tumor-bearing mice were performed for histological analyses (Fig. 5m and Supplementary Fig. 51). Clearly, carcinoma (the abnormal cells with darker, larger, and irregular nuclei in H&E image) has invaded the bladder wall of mice 20 days post T24 cell inoculation. Moreover, some bladder tissues of healthy and tumor-bearing mouse were stained with fluorescently labeled CD44-antibody, and their typical immunofluorescence (IF) images are shown in Fig. 5n. It can be found in the figure that, the expression level of CD44 (displayed in red) is much higher in the bladder tissue of the mouse 20 days upon intravesical inoculation with T24 cancer cells compared

to that in normal bladder from the healthy mouse (control). These results further validate that the capability of the nanoagent DXBTZ-CB[8]/CSA to detect and localize orthotopic bladder tumor in vivo by targeting the overexpressed CD44 receptors via MSOT imaging.

## Multispectral optoacoustic imaging of lymphatic metastasis of tumor in a mouse model using the nanoagent DXBTZ-CB[8]/CSA

As one of the spreading pathways for cancer, lymphatic metastasis would allow tumor cells to be transported from primary site through lymphatic vessels to closest regional lymph node (also called sentinel lymph node), ultimately to other tissues of the body[55]. Since local spread of tumor from the primary site to a regional lymph node often represents a sign of poor prognosis, accurate detection of metastases on the sentinel lymph node is important for formulating an appropriate treatment plan. In this experiment, we utilized DXBTZ-CB[8]/CSA to track the lymphatic metastasis in a mouse model after the primary tumor formation by injecting 4T1 cells (mouse mammary adenocarcinoma cell line) into the footpad (Fig. 6a). As shown in Fig. 6b, the obtained cross-sectional images present the merging of DXBTZ-CB[8]/CSA signal on the anatomical reference shown in grayscale background and the MSOT signal of DXBTZ-CB[8]/CSA alone, respectively. Moreover, the quantified MSOT signal intensities of the nanoagent DXBTZ-CB[8]/CSA at the region of interest are given in Fig. 6c. In addition, key tissues (femur and artery) in that cross section were marked for comparison with the cryosection image for the popliteal fossa of female mice (Fig. 6d). As displayed in Fig. 6b, c, the MSOT signal of DXBTZ-CB[8]/CSA at the popliteal lymph node site could be lucidly visualized on day 21 after inoculation of 4T1 cancer cells in the footpad, implying that the tumor cells metastasize into the regional lymph node.

In addition, other cross-sectional MSOT images (e.g., primary tumor site at the footpad in Supplementary Fig. 52) corresponding to the positions from hind feet to popliteal fossa of the mice 21 days post inoculation of 4T1 cancer cells were also recorded to generate the orthogonal MIP MSOT images. As shown in Fig. 6e, the MSOT signal of DXBTZ-CB[8]/CSA in the obtained orthogonal MIP image evidently maps the lymphatic spread pathway from primary site (footpad) via lymphatic vessels to popliteal lymph node. Afterwards, the mice were euthanized with carbon dioxide, and their skin at the popliteal fossa site was removed. The popliteal site and footpad site of mice from different groups were photographed and displayed in Fig. 6f and Supplementary Fig. 52. As compared with the popliteal lymph node from control mouse, the one of mouse inoculated with 4T1 cells is significantly enlarged, indicating the invasion by metastatic tumor cells (Fig. 6f). Moreover, the popliteal lymph node of tumor-bearing mouse at 1 h after

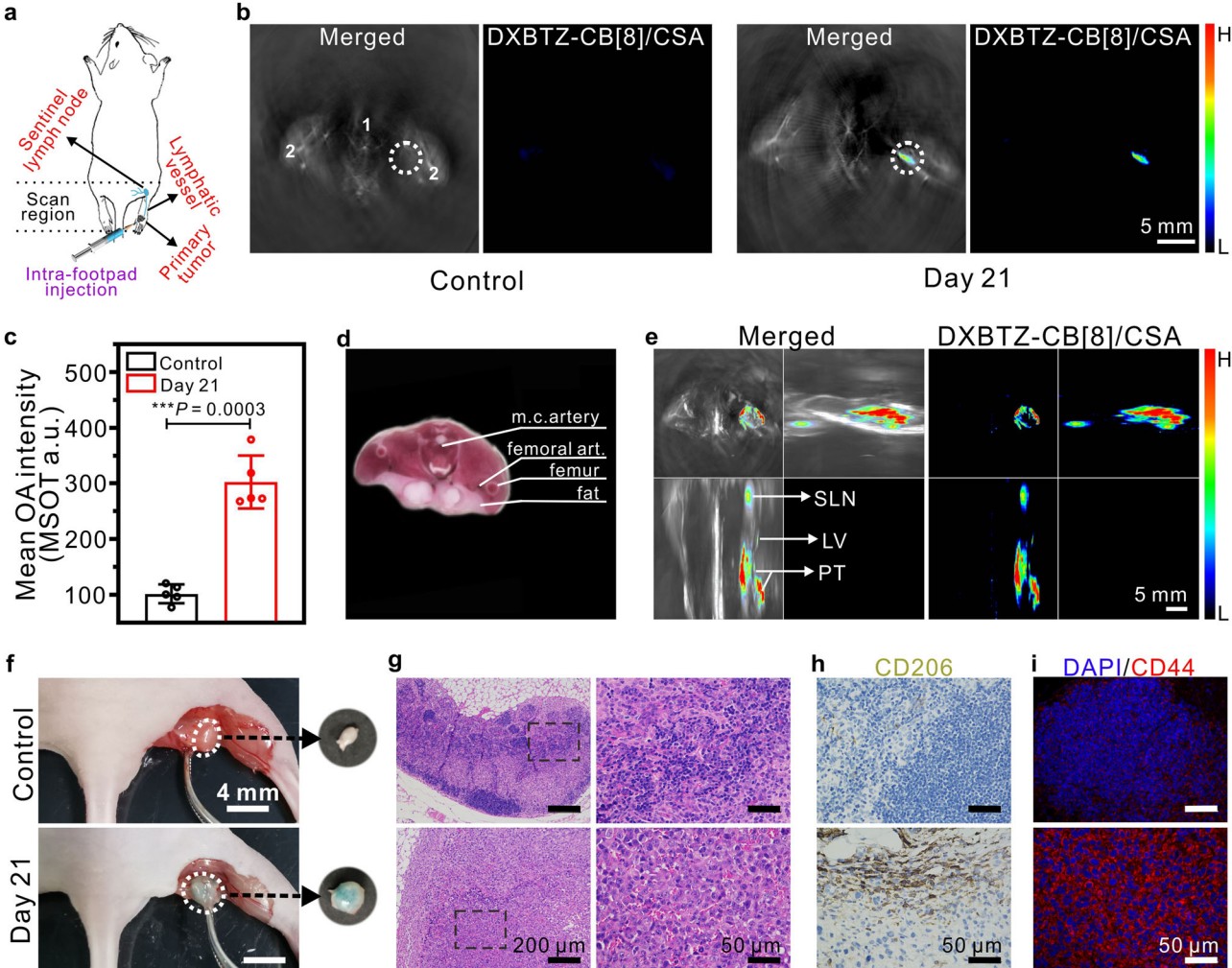

**Fig. 6 | Application of DXBTZ-CB[8]/CSA in the mouse model of tumor lymphatic metastasis. a** Schematic illustration for multispectral optoacoustic imaging of lymphatic metastasis of tumor in mice after intra-footpad injection of DXBTZ-CB[8]/CSA. **b** Typical cross-sectional MSOT images corresponding to the popliteal fossa position of the control group and the tumor lymphatic metastasis model group. The mice were placed in the prone position for imaging. Left panel: overlay of DXBTZ-CB[8]/CSA optoacoustic signal onto the grayscale image with anatomical information. Right panel: spectrally unmixed optoacoustic signal of DXBTZ-CB[8]/CSA. Organ tags: 1: artery; 2: femur. The popliteal lymph node region of mice was delimited with a white dotted circle (ROI). **c** Mean MSOT intensities at ROI for the control mice and the mice with tumor lymphatic metastasis at 1 h post intra-footpad administration of DXBTZ-CB[8]/CSA (31.1 mg kg$^{-1}$; $n = 5$ animals per group). **d** A cryosection image of a female mouse corresponding to the cross-section location in **b**. **e** Orthogonal MIP MSOT images for the tumor lymphatic metastasis model mice. The primary tumor (PT), lymphatic vessel (LV) and sentinel lymph node (SLN) of the model mouse were pointed by white arrows in the image. **f** Photos for the popliteal lymph node (before and after resection) of the control mouse and the mouse (with lymphatic metastasis) 1 h post intra-footpad injection of DXBTZ-CB[8]/CSA. **g** Typical H&E-stained sections for the popliteal lymph node from the control mouse and the mouse (with lymphatic metastasis) 21 days post intra-footpad inoculation of 4T1 cells. The right panel displays the magnified H&E images of the areas in the black dotted box in left panel. **h** Immunohistochemistry images of anti-CD206-stained popliteal lymph node sections from the control mouse and the mouse 21 days post intra-footpad inoculation of 4T1 cells. **i** Immunofluorescence images of anti-CD44-stained popliteal lymph node sections from the control mouse and the mouse 21 days post intra-footpad inoculation of 4T1 cells. Scale bar: 50 μm. Data are presented as mean values ± SD. Statistical significance was determined by two-tailed $t$ test. ***$P < 0.001$. Source data are provided as a Source Data file.

intra-footpad injection of DXBTZ-CB[8]/CSA displays in greenish blue color, suggesting that the nanoagent can track the metastatic cancer cells in lymphatic system via CD44 targeting. Also, it can be seen from Supplementary Fig. 52 that the right hind foot of the mouse is swollen, indicating that the injected 4T1 cells form the primary tumor at paw. In contrast, the control mice didn't exhibit these symptoms. These observations confirm that the CD44-targeted nanoagent DXBTZ-CB[8]/CSA could diagnose lymphatic metastasis of tumors and precisely map the entire metastasis pathway via MSOT imaging.

To further validate the lymphatic metastasis, the histopathological analysis for the popliteal lymph node of the control and tumor-bearing mice was performed by H&E staining. As displayed in Fig. 6g, the irregularly shaped cells with large nuclei in the H&E section

represent the presence of tumor cells in the lymph node of mice 21 days post 4T1 cell inoculation. Moreover, the expression of macrophage mannose receptor (CD206) in the sentinel lymph node was evaluated by immunohistochemical (IHC) staining. As an important indicator for cancer diagnosis, CD206 overexpression has been found in tumor-associated macrophages (TAM) to perform a vital role in tumor proliferation, invasion, and metastasis[56]. It can be clearly observed from Fig. 6h that, 21 days post intra-footpad inoculation of 4T1 cells, the mice expressed much higher CD206 receptors in the sentinel lymph node than the control ones. This observation reveals that the cancer cells spread from the injection (primary) site to the popliteal lymph node, forming the metastatic tumor in mice. Furthermore, the CD44 level in the lymph nodes was also determined by

immunofluorescence staining. As shown in Fig. 6i, the popliteal lymph node of tumor-bearing mice expresses significantly more CD44 receptors compared to that of control mice, thus providing sufficient targeting sites for the nanoagent DXBTZ-CB[8]/CSA to achieve MSOT imaging and diagnosis of lymphatic metastasis. These analyses buttress our imaging results, and further demonstrate that the nanoagent DXBTZ-CB[8]/CSA is able to identify lymphatic metastasis of tumors by means of multispectral optoacoustic imaging.

### Multispectral optoacoustic imaging of renal I/R injury in a mouse model using the nanoagent DXBTZ-CB[8]/CSA

In addition to tumor diagnosis, application of the nanoagent DXBTZ-CB[8]/CSA in detection of inflammatory diseases was also explored. As a health concern of global significance, renal I/R injury is primarily caused by kidney transplantation, hemorrhagic shock as well as prolonged resuscitation, and manifests as a sudden loss of renal function[57]. Currently, the clinical diagnostic method mainly relies on the determination of blood urea nitrogen (BUN) and serum creatinine (CREA), which are typically considered to be abnormally elevated due to the decrease in estimated glomerular filtration rate during renal I/R injury[58]. However, BUN and CREA can also be regulated by numerous kidney-independent factors (e.g., drug intake, protein ingestion, and catabolic state), which would greatly reduce the accuracy of diagnosis[59]. Given the rising morbidity and high mortality rate, there is an urgent need to develop a more intuitive method for detection of I/R-induced acute kidney injury (AKI), which facilitates timely protective interventions for kidney recovery. In this study, we generated a mouse model of I/R-induced AKI according to the commonly-used method that the bilateral renal pedicles were isolated and clamped for 60 min to induce ischemia, followed by reperfusion for 24 h[60]. Upon renal injury, expression of CD44 receptors is significantly upregulated on the tubuloepithelial cells[61,62]. In addition, AKI also leads to enhanced permeability of inflamed renal microvasculature, which can promote the accumulation of nanoagents in the kidney[63,64]. Photographs for our detailed modeling procedures of renal I/R injury are displayed in Fig. 7a.

As for MSOT imaging, the sham-operated control mice as well as the I/R-induced AKI mice were administered with DXBTZ-CB[8]/CSA via intravenous injection (Fig. 7b), and the obtained cross-sectional images are presented in Fig. 7c. Mean MSOT intensity in the ROI corresponding to the kidney tissue of mice is displayed in Fig. 7d, which offers intuitive data for the cross-sectional images. It can be clearly seen that the strong MSOT signal of DXBTZ-CB[8] are mainly localized in the kidneys of the AKI mouse by referring to the cryosection reference (Fig. 7e), while the control mouse only exhibits a weak MSOT signal in the kidney region. In addition, we also produced the orthogonal MIP MSOT image for AKI mice through z-stack rendering. From the resulted orthogonal image (Fig. 7f), the injured bilateral kidneys can be clearly visualized.

After the mice were killed, the main organs were resected and then subjected to MSOT imaging. As revealed from the ex vivo imaging results in Fig. 7g, h, the kidneys of AKI mouse display a much more prominent MSOT signal than other organs, which validate the in vivo imaging observations. Moreover, plasma BUN and CREA levels were measured to assess the renal function in mice. Compared with the control, the mice receiving ischemic exposure demonstrated a distinct exaltation in the plasma levels of BUN and CREA, implying the development of kidney dysfunction (Fig. 7i). In addition, H&E staining, immunohistochemistry, and immunofluorescence analyses were also conducted on the renal sections of the mice that underwent sham surgery or I/R procedure. Compared to the sham-operated control mice, the AKI mice exhibited abnormal dilation, distortion, and even degeneration of renal tubules (Fig. 7j). As shown in Fig. 7k, the AKI mice expressed markedly higher levels of Interleukin-6 (IL-6, a pro-inflammatory cytokine) in the kidneys than the control mice, proving inflammatory kidney injury post ischemic exposure. Meanwhile, the

expression of CD44 (displayed in red color) receptors in the kidney of the mice with renal I/R injury is also much higher than that of the control mice (Fig. 7l). The confirmation of enhanced renal CD44 expression post renal I/R injury provides a scientific basis for targeted imaging of damaged kidneys with DXBTZ-CB[8]/CSA.

Afterwards, we performed western blot experiments to examine the level of matrix metalloproteinase (MMP-2 and MMP-9) in kidney tissue harvested from the mice in different groups. Previous studies have indicated that upregulation of MMP-2 and MMP-9 leads to increased proteolysis of the perivascular matrix such as collagen IV, thereby enhancing the microvascular permeability in tissues[63,65]. As seen in Fig. 7m and Supplementary Fig. 53, elevated expression of MMP-2 and MMP-9 was detected in the damaged kidneys from the AKI mice, suggesting increased renal microvascular permeability following ischemic exposure. The enhancement of renal microvascular permeability favors the accumulation of DXBTZ-CB[8]/CSA and subsequent elicitation of strong MSOT signal at injured kidney sites of the AKI mice upon photoexcitation. All these pathological findings fully sustain our MSOT imaging observations, and further illustrate that the CD44-targeted nanoagent DXBTZ-CB[8]/CSA can effectively detect and image renal I/R injury owing to overexpression of CD44 receptors and enhanced microvascular permeability in damaged kidneys.

## Discussion

Organic small-molecule chromophores as promising contrast agents for multispectral optoacoustic imaging often suffer from the impediments including relatively low extinction coefficients and poor water solubility/dispersibility, thus restricting their extensive applications in biomedical fields. In this study, we have demonstrated that these hurdles can be circumvented by constructing water-dispersible supramolecular optoacoustic assemblies based on cucurbit[8]uril.

In summary, we have synthesized two organic model chromophores (DXP and DXBTZ) as guest compounds, and then constructed water-dispersible supramolecular complexes via the host–guest interactions between macrocyclic cucurbit[8]uril and the synthetic dixanthene derivatives. Compared with the chromophore DXP or DXBTZ alone, the resulting supramolecular host–guest complexes (DXP-CB[8] and DXBTZ-CB[8]) exhibit a significant enhancement in the optoacoustic signal upon excitation, which is attributed to the increased absorption coefficients and intensified non-radiative relaxation of the excited state by suppressing the competitive radiative transition. In particular, DXBTZ-CB[8] displays a characteristic red-shifted absorption peak at 692 nm within the detection range (680–980 nm) of the multispectral optoacoustic imaging system, which contributes to definite identification in spectral unmixing, and thus being further functionalized into a CD44-targeted nanoagent via co-assembly with CSA. The formulated ternary nanoagent DXBTZ-CB[8]/CSA not only maintains the unique spectral characteristics and excellent optoacoustic performance, similar to the binary supramolecular complex DXBTZ-CB[8], but also can efficiently target and accumulate in tumor cells or diseased cells with CD44 overexpression. With multispectral optoacoustic imaging, DXBTZ-CB[8]/CSA is able to detect and visualize subcutaneous tumors, orthotopic bladder tumors, lymphatic metastasis of tumors and renal I/R injury in mouse models. Furthermore, the generated orthogonal MIP MSOT images can be utilized to accurately localize the disease foci. This study herein serves as the proof of concept for constructing CB[8]-based supramolecular assemblies to enhance the optoacoustic performance of organic contrast agents by improving their extinction coefficients as well as water dispersibility and adjusting the deactivation pathways of their excited state, and demonstrates their promising potential in biomedical applications with the aid of multispectral optoacoustic imaging. Moreover, since CB[8] could extend the conjugation of guest chromophores upon complexation, a great number of existing organic chromophores including those with absorption only at wavelengths

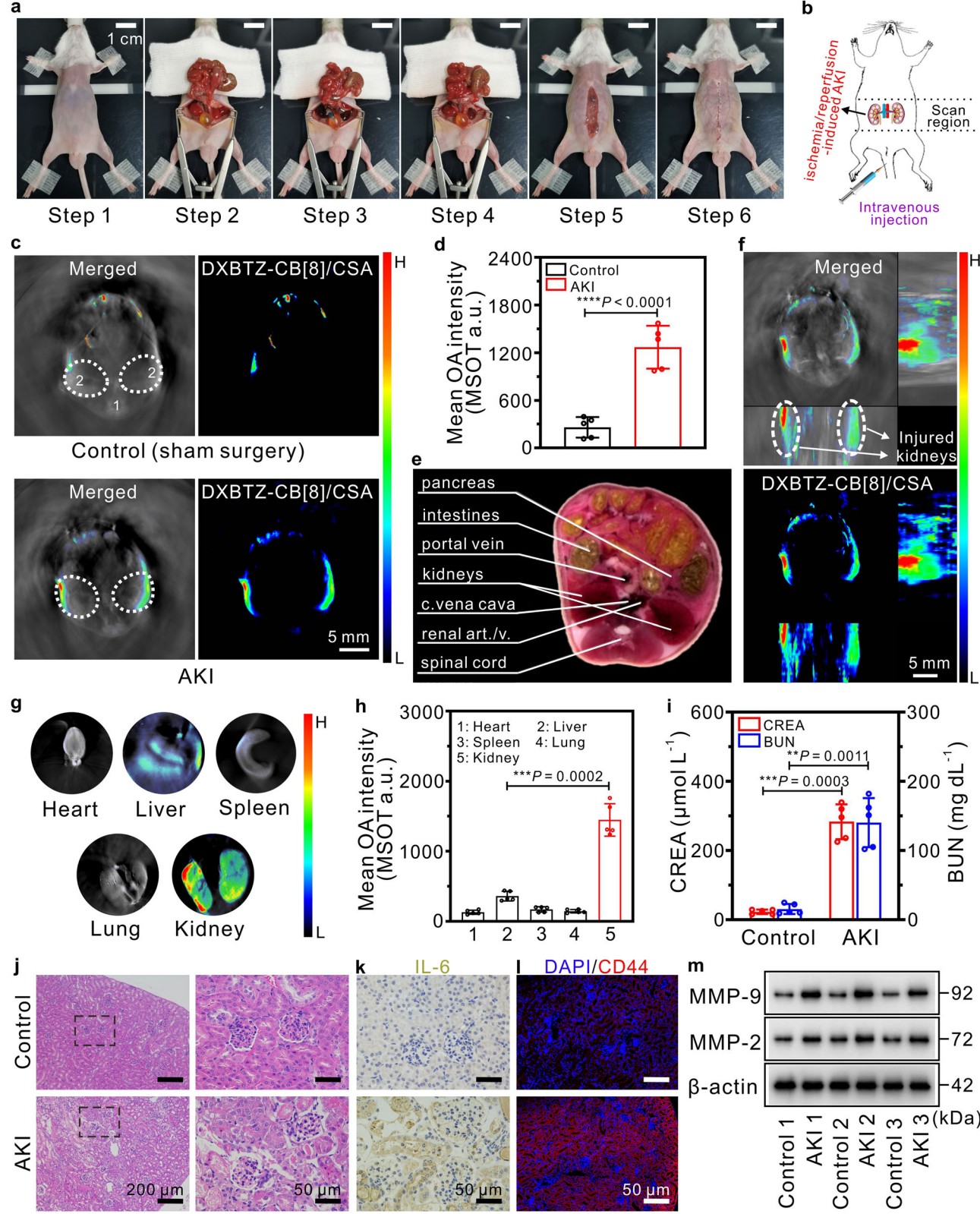

less than 680 nm, can be exploited this way to fabricate robust optoacoustic contrast agents for more extensive applications.

## Methods

### Determination of binding stoichiometry and association constant (Ks)

In Job's plot experiments, the total concentration of the guest molecule DXP or DXBTZ and host molecule CB[8] was fixed at 10 μM. The

UV absorption spectrum was recorded by changing guest molecule DXP or DXBTZ molar ratio from 0.1 to 0.9 where the CB[8] molar ratio synchronously changed from 0.9 to 0.1. In the control experiments, the absorption of the same concentration of guest molecule DXP or DXBTZ solutions without CB[8] were also measured. The absorbance subtraction at 730 nm was recorded to calculate the binding ratio for DXP and CB[8]. Similarly, the absorbance subtraction at 803 nm was recorded to determine the binding stoichiometry of DXBTZ and CB[8].

**Fig. 7 | Application of DXBTZ-CB[8]/CSA in the mouse model of renal I/R injury. a** Photographs for the surgical process of I/R-induced acute kidney injury in mouse model. **b** Schematic illustration for multispectral optoacoustic imaging of I/R-induced AKI in mice after intravenous injection of DXBTZ-CB[8]/CSA. **c** Typical cross-sectional MSOT images for the bilateral kidney of the control mouse and the AKI mouse 5 h post intravenous injection of DXBTZ-CB[8]/CSA dispersion. Left panel: overlay of DXBTZ-CB[8]/CSA optoacoustic signal onto the grayscale image with anatomical information. Right panel: spectrally unmixed optoacoustic signal of DXBTZ-CB[8]/CSA. Organ tags: 1: spinal cord; 2: kidney. The bilateral kidneys of mice were delimited with two white dotted circles (ROI). **d** Mean MSOT intensities at ROI for the control group and the AKI group at 5 h post intravenous administration of DXBTZ-CB[8]/CSA (62.2 mg kg$^{-1}$; $n = 5$ animals per group). **e.** Cryosection image of a male mouse corresponding to the cross-section location in **c**. **f** Orthogonal MIP MSOT image for the renal I/R injury mouse model. **g** Typical MSOT images for the major organs excised from an AKI mouse 5 h post intravenous injection of DXBTZ-CB[8]/CSA. **h** Mean MSOT intensities of the major organs at 5 h upon injection of DXBTZ-CB[8]/CSA ($n = 5$ animals per group). **i** Levels of CREA and BUN in blood of the control mice and AKI mice ($n = 5$ animals per group). **j** Typical H&E-stained sections for the kidney from the control mouse and the AKI mouse. The right panel displays the magnified H&E images of the areas in the black dotted box in left panel. **k** Immunohistochemistry images of anti-IL-6-stained kidney sections from the control mouse and the AKI mouse. **l** Immunofluorescence images of anti-CD44-stained kidney sections from the control mouse and the AKI mouse. **m** Representative western blotting analysis illustrating the level of MMP-2 as well as MMP-9 in the kidney of the control mice and AKI mice. Data are presented as mean values ± SD. Statistical significance was determined by two-tailed $t$ test. **$P < 0.01$, ***$P < 0.001$ and ****$P < 0.0001$. Source data are provided as a Source Data file.

For determination of the association constant of guest molecule DXP or DXBTZ and CB[8], the UV-vis titration experiment was carried out, in which the concentration of guest molecule was constant and the concentration of host molecule CB[8] varied. The binding constant was obtained from a 1:1 binding model by using the non-linear least-squares fit of the variation of UV absorbance at 658 nm for DXP and at 692 nm for DXBTZ.

## Preparation of the binary host–guest complex DXP-CB[8] and DXBTZ-CB[8]

Briefly, an equimolar amount of guest molecule DXP or DXBTZ (0.02 mmol) and host molecule CB[8] (0.02 mmol) was mixed in water (10 mL), followed by sonication and stirring at room temperature to generate the binary host–guest complex. For DXBTZ-CB[8] preparation, the guest molecule DXBTZ was completely dissolved in a small amount of DMSO before complexing with CB[8], and the final DXBTZ-CB[8] solution contained 10% DMSO (v/v). Both the binary host–guest complex DXP-CB[8] and DXBTZ-CB[8] solution were stored at room temperature in the dark.

## Preparation of the ternary nanoagent DXBTZ-CB[8]/CSA

CSA (16 mg) was added into the above-prepared DXBTZ-CB[8] complex solution. After 30 min gently shaking, the obtained CD44-targeting nanoagent DXBTZ-CB[8]/CSA was dialyzed against water (molecular weight cut-off 500 Da) for 48 h to remove DMSO followed by lyophilization. The dried nanoagent DXBTZ-CB[8]/CSA was resuspended in water (10 mL), shaken gently, and used directly for further experiments. The concentration of DXBTZ-CB[8]/CSA stock solution was calculated to be 5.6 mg/mL, containing 2 mM of DXBTZ-CB[8] complex.

## Multispectral optoacoustic imaging

All in vitro phantom and in vivo mouse multispectral optoacoustic experiments were conducted on the inVision 128 MSOT device. For experiments in phantom, the test solutions were filled in NMR tubes, and then mounted on the device holder. The optoacoustic images of DXP mixed with different concentrations of CB[8] were acquired 680 nm, while those of DXBTZ mixed with different concentrations of CB[8] were collected at 692 nm. Optoacoustic images of the nanoagent DXBTZ-CB[8]/CSA of varied concentrations were also obtained using 692 nm as the excitation wavelength. To compare the stability, the same mass concentration of DXBTZ-CB[8]/CSA and commercially available gold nanorods were exposed to 10000 consecutive laser pulses at 692 nm, and 10 average frames was set up for each repetition.

For in-vivo multispectral optoacoustic imaging experiments of subcutaneous MDA-MB-231 tumor model, the mice were anesthetized by 1% isoflurane delivered via a nose cone, and then intravenously injected with the nanoagent DXBTZ-CB[8]/CSA dispersion (62.2 mg kg$^{-1}$). The tumor region of each mouse was imaged at pre-determined time intervals (0, 3, 6, 9, 12, 24, 48, and 72 h). The feces and urine of these mice were also collected at 0.5, 1, 3, 5, and 7 days after administration of DXBTZ-CB[8]/CSA. The optoacoustic intensity of these excreta solutions was acquired upon excitation at 692 nm and the cumulative clearance percentage was calculated from the standard curve. As for the orthotopic bladder tumor model, the mice were intravesically injected with DXBTZ-CB[8]/CSA dispersion (15.6 mg kg$^{-1}$). Upon 1 h post administration, the dispersion was drawn back into the syringe, the catheter was withdrawn, and then the mice were imaged in the prone position. For in vivo MSOT studies of lymphatic metastasis model, mice was given DXBTZ-CB[8]/CSA dispersion (31.1 mg kg$^{-1}$) via intra-footpad injection, and MSOT imaging was conducted 1 h post administration. As for renal I/R injury model, the AKI mice were intravenously injected with DXBTZ-CB[8]/CSA dispersion (62.2 mg kg$^{-1}$). After 5 h, these mice were placed in the supine position, and then scanned to acquire MSOT images. During the data collection process, 680 nm, 692 nm, 700 nm, 730 nm, 760 nm, 800 nm, 850 nm (background), and 900 nm were selected considering the major inflection points corresponding to the absorption curves of the nanoagent DXBTZ-CB[8]/CSA, oxyhemoglobin, and deoxyhemoglobin, and 10 individual frames were acquired for each wavelength. Subsequently, spectral unmixing was performed by using a linear regression algorithm to extract optoacoustic signals generated from DXBTZ-CB[8]/CSA. Orthogonal MIP MSOT images were generated by stacking cross-sectional images along the z-axis. Cryosection images provided with *viewMSOT* software in the MSOT system were used for the reference to anatomical details. The maximal tumor size/burden of 1000 mm$^3$ was permitted by the ethics committees and the maximal tumor size/burden in this study was not exceeded.

## Normalized optoacoustic intensity

It was calculated according to the equations (1) or (2):

$$\text{For DXP, Normalized optoacoustic (OA) intensity} = (OA_{680})_{\text{DXP-CB[8]}}/(OA_{680})_{\text{DXP}} \tag{1}$$

$$\text{For DXBTZ, Normalized OA intensity} = (OA_{692})_{\text{DXBTZ-CB[8]}}/(OA_{692})_{\text{DXBTZ}} \tag{2}$$

## Ethics statement

All animal experiments were approved and conducted in compliance with the regulations of Ethics Committee of Laboratory Animal Center of South China Agricultural University (Approval No. 2020d076) and the Guidelines for Care and Use of Laboratory Animals of the Institutional Animal Care and Use Committee of Nanyang Technological University (NTU-IACUC) (Approval No. A19016).

## Statistics and reproducibility

All experiments were repeated independently at least three times with similar results. Quantitative data were expressed as mean ± standard deviation (SD). NMR spectra were analyzed by using MestreNova LITE

v14.0.0.0-23239 software and JEOL Delta v6.1.0 software. All the simulations were performed by using the Gaussian 16_A01 program package, and the HOMO and LUMO plots were visualized using VMD 1.9.3 software. Flow cytometry results were analyzed by Flow Jo v10.0.7. Data analysis of imaging was done using viewMSOT 3.8 and the Living Image 4.3 software. Statistical calculations and data analysis were performed using OriginPro and GraphPad Prism software. The statistical comparisons were conducted by two-tailed unpaired Student's $t$ test. $P$ values less than 0.05 were considered significant. $*P < 0.05$, $**P < 0.01$, $***P < 0.001$, $****P < 0.0001$.

### Reporting summary

Further information on research design is available in the Nature Portfolio Reporting Summary linked to this article.

## Data availability

The authors declare that all the data supporting the findings of this study are available within the article and its Supplementary Information, and the full image dataset is available from the corresponding author upon request. Source data are provided with this paper. A reporting summary for this article is also available as a Supplementary Information file. Source data are provided with this paper.

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

## Acknowledgements

The work was supported by the Singapore Agency for Science, Technology and Research (A*STAR) Advanced Manufacturing and Engineering (AME) Individual Research Grant (No. A20E5c0081 to Y.Z.) and the Singapore National Research Foundation Investigatorship (No. NRF-NRFI2018-03 to Y.Z.). This work was also supported by the National Natural Science Foundation of China (No. 21788102 to S.W. and No. 32101065 to H.C.) and the Fund of Guangdong Provincial Key Laboratory of Luminescence from Molecular Aggregates (No. 2019B030301003 to S.W.).

## Author contributions

Y.Z., L.S., S.W., H.C., F.Z., and Y.W. conceived the project. Y.W. designed the chemical synthetic route. Y.W., L.S., X.C., J.L., J.O., X.Z., Y.G., Y.C., W.Y., D.W., and T.H. conducted the experiments. Y.Z., L.S., F.Z., S.W., H.C., and Y.W. wrote the manuscript. All the other co-authors contributed to data interpretation and revision of manuscript.

## Competing interests

The authors declare no competing interests.
