## [Peer Review File · Nature Communications]

Reviewers' Comments:

Reviewer #1:

Remarks to the Author:

The manuscript with reference number NCOMMS-22-25756A-Z entitled " Cucurbit[8]uril-Based Water-Soluble Assemblies with Enhanced Optoacoustic Performance for Multispectral Optoacoustic Imaging" used supramolecular assemblies to solve the solubility and improve the extinction coefficient of organic molecules, and improved the biocompatibility and specificity by the introduction of targeted polysaccharide chondroitin sulfate A. The overall work is systematic and practical, but the interactions of supramolecular assemblers is not deep enough and needs to be improved.

(1) When discussing the interaction modes (J-aggregate or H-aggregation) of assemblers, it is not enough to prove the hypothesis only by ultraviolet spectral redshift. Other more intuitive evidence should be provided, such as NOESY.

(2) The ¹H NMR titration of host and guest is one of the best methods in studying the host-guest interaction in the absence of crystal structure. However, this paper did not analyze them in detail, and only described the improvement of water solubility. Obviously, the addition of the melon ring does not only increase the water solubility but also the shift of the proton signal, so the proper assignment of the proton signal facilitates the demonstration of the interaction modes in Figure 1.

(3) The authors stated that the assembly of DXBTZ-CB[8] and CSA does not affect the extinction coefficient of DXBTZ-CB[8]. However, no UV and fluorescence spectral changes were found, what is the evidence for their assembly? Can TEM alone prove that these assemblies are highly stable and do not dissociate in the organism?

(4) What is the driving force of the assembly, electrostatic interactions or hydrogen bonding?

(5) Fig 4g and 4j prove that DXBTZ-CB[8]/CSA only enters the cell through highly expressed CD44 receptors on the cell surface, which is insufficient. In Fig 4g, there is no red channel in HEK293 cells, indicating that DXBTZ-CB[8]/CSA cannot enter the cell, but the reasons why the material fails to enter the cell may be diverse. Other cells that do not express CD44 should be supplemented as a reference comparison. In Fig 4j, HA is used as a competitive inhibitor, but in addition to CD44, HA may also bind to other proteins on the cell surface to block DXBTZ-CB[8]/CSA from entering the cell.

(6) In Fig 5h and Fig 6b, DXBTZ-CB[8]/CSA enters the lesion site through in situ injection. Why did the photoacoustic imaging of DXBTZ-CB[8]/CSA disappear in the blank control group after just an hour? Is it metabolism? In Fig 5b, the material does not attenuate until 12 hours after entering the body.

(7) In Fig 7d, DXBTZ-CB[8]/CSA is heavily enriched in the AKI model (injection concentration of 62 mg/kg). At the same time, in Fig 5f, this material is also widely enriched in the kidneys (injection concentration of 46.7 mg/kg) and tumor sites. Will the difference between the data in Fig 7d and Fig 5f be caused by the different concentration of injection?

Other minor questions:

(1) The diffusion coefficient is an important data for supramolecular assemblers (DXBTZ-CB[8] and DXP-CB[8])

(2) The content of Figure 2e is inconsistent with the description in the text.

(3) How does the assemblies metabolic clearance from the tumor?

Reviewer #2:

Remarks to the Author:

Multispectral optoacoustic imaging is an interesting approach to promote the spatial resolution and tissue penetration depth and has a major impact on the field of preclinical and clinical applications. In this manuscript, the authors reported two water-soluble supramolecular assemblies based on cucurbit[8]uril and validated their utility in several animal models. The constructed CB[8]-based assemblies exhibited significantly enhanced optoacoustic performance compared to the individual guest organic small-molecule chromophore. After being coated with targeting components, the assemblies successfully achieved the diagnosis and localization of subcutaneous tumor, orthotopic bladder tumor, lymphatic metastasis of tumor and renal ischemia/reperfusion injury in mice by

multispectral optoacoustic imaging. Overall, this work is rather interesting and is well presented with proper discussions.

1. Statistical analysis of the data should be provided, and this should be added especially for the data in Fig. 5-7.
2. The source of cryosection images shown in Fig. 5j, 6d, 7e and Supplementary Fig. 36b for reference to anatomical details should be stated in the Methods Section.
3. In Supplementary Fig. 23, the optoacoustic spectrum (a or b) should be clearly assigned to DXP (or DXBTZ)-CB[8] assemblies.
4. In Supplementary Fig. 34, the format of horizontal axis labels should be optimized for readability.

Reviewer #3:

Remarks to the Author:

In this manuscript, the supramolecular complexes based on the host-guest chemistry of cucurbit[8]uril and dixanthene derivative were developed for multispectral optoacoustic imaging of tumors and so forth. The selling point of this work should result from water solubility of assemblies, that is, the DXBTZ-CB[8]. Unfortunately, the assemblies and nanoparticles with CD44-targeting capability are dispersed solid systems, which compromises the novelty and significance of this present work. Also, the introduction of targeting molecules (CSA) on the drug nanoparticles is not a new concept. On the whole, the approach and material property delivered in this work have been broadly reported. It can only be considered an extension of the cucurbit[8]uril host-guest chemistry for preparation of dixanthene derivative contrast agents for imaging application. Therefore, rejection should be recommended and the following questions are suggested consideration for submission elsewhere.

Major points

- Cucurbit[8]uril has been broadly used as the building block for host-guest supramolecular chemistry. The incorporated contrast agents into CB[8] can be easily imagined. Therefore, it is the overstatement of the strategy reported in this work regarded as the newly-developed easy-to-follow approach.
- Multispectral optoacoustic imaging technique shows the advantageous in comparison to the other optical imaging ones, but it is unclear how the imaging agents impede or facilitate its applications preclinically or clinically. In other words, how can the development of photoacoustic contrast agents fasten the clinical applications of multispectral optoacoustic imaging technique?
- The authors stated "physical encapsulation often leads to broadened as well as decreased absorption peak(s) and subsequent reduction in optoacoustic signal due to the random aggregation of contrast agents." The reviewer can not follow the translation of reduction in acoustic signal as the random aggregation of contrast agents. One can well understand the reduction of fluorescent signal is due to the aggregation induced quenching, but not for photoacoustic agents. They share different the mechanism with that the excited state down to the ground state in different pathway. The thermal relaxation is not subjected to the aggregation of molecular chromophores, oppositely, the aggregation may enhance the acoustic signal due to the blocking of other energy pathways.
- No experimental methods were showed on how the nanorods were prepared and corresponding conditions. It is unclear why the resulting assemblies are the rods, but not other structures. No experimental data and discussion on the storing stability that are of key importance for supramolecular nanodrugs. Do the nanorods evolve over aging time, e.g. months or longer? What maintains the stability of nanorods?
- It is inappropriate to state the assemblies can be water soluble as either nanorods and nanoparticles are clearly dispersed solid system, same to all the nanodrugs dispersed in water reported previously. It is suggested to clearly distinguish the solid dispersion system from water soluble system. They are completely different phases.

- It is confusing to follow the morphological transformation from nanorods to nanoparticles after mixing with CSA. How to ensure the stable host-guest complexation between dixanthene derivative and CB[8] in the presence of CSA? The dixanthene derivative and CB[8] assemblies should be nanorods, looking like a thermodynamically favorable crystal. In principle, the coating can not change the inherent nanorod morphology of binary complexes. How does CSA alter the morphology of resulting assemblies? The morphological changes hint the multiple interactions of CSA, dixanthene derivative and CB[8]. What is the mechanism?
- In the tumor imaging sections, one can not follow how well the nanoparticles of DXBTZ-CB[8] complexes for imaging the tumors as the positive controls with other contrast agents such as gold nanoparticles are missing.

Minor points

- How to experimentally determine all the resulting complexes being binary, but not mixture of various complexes? The direct evidence of binary complexes is highly appreciated.
- How was polysaccharide chondroitin sulfate A (CSA) coated on DXBTZ-CB[8]?
- What is the solubility of the two positively charged chromophores, dixanthene derivatives (DXP and DXBTZ) in water?
- The absorption less than 700 nm can not be considered near-infrared absorption. Red light absorption should be more accurate.
- In Supplementary Figure 29, the stability of nanoparticles was only evaluated for 24h at most. This is insufficient and the extended aging time is encouraged.
- In Figure 4c, the complete overlapping of red and green curves is questionable. This should be double checked. The turbidity in the two cases should be quite different. Thus, it is impossible to have two completely overlapped absorption curves.
- The details on the measurement of flow cytometry for verifying the uptake of nanoparticles should be supplemented. What is responsible for the fluorescence of cells after uptake of nanoparticles? Does this mean the decomplexation of binary complexes and release of molecular contrast agents?

Response to Reviewers' Comments

Reviewer #1 (Remarks to the Author):

The manuscript with reference number NCOMMS-22-25756A-Z entitled " Cucurbit[8]uril-Based Water-Soluble Assemblies with Enhanced Optoacoustic Performance for Multispectral Optoacoustic Imaging" used supramolecular assemblies to solve the solubility and improve the extinction coefficient of organic molecules, and improved the biocompatibility and specificity by the introduction of targeted polysaccharide chondroitin sulfate A. The overall work is systematic and practical, but the interactions of supramolecular assemblers is not deep enough and needs to be improved.

Our Response: We thank the reviewer for the useful comments and advice. We have revised the manuscript accordingly.

(1) When discussing the interaction modes (J-aggregate or H-aggregation) of assemblers, it is not enough to prove the hypothesis only by ultraviolet spectral redshift. Other more intuitive evidence should be provided, such as NOESY.

Our Response: We thank the reviewer for his/her comments and suggestions. In this revised manuscript, several 2D NMR experiments (including NOESY, DQF-COSY, HSQC and DOSY) have been performed to prove the hypothesis, and the results are displayed in Supplementary Figure 30-41. For DXBTZ, the aromatic Proton 4 and methyl group Proton 11 having a distance larger than 5 Å in a single molecule produced the NOE effect after complexing with CB[8]. The cross peak as the result of reduced distance between Proton 4 and 11 in the NOESY spectrum of DXBTZ-CB[8] indicated that DXBTZ molecule could only adopt the J-aggregation interaction mode in the CB[8] cavity (Supplementary Figure 40). Moreover, in Supplementary Figure 38, a significant upfield shift of Proton 11' peak was observed due to the shielding effect of the resulted J-aggregates, which could also provide evidence to verify the hypothesis (*Angew. Chem. Int. Ed.*, 2023, **62**, e202216109; *Angew. Chem. Int. Ed.*, 2016, **55**, 10093-10097). As for DXP, although the distance between the aliphatic chain Proton 7 or 11 and the alkene moiety Proton 2 is much larger than 5 Å in a single molecule, no cross peak between them was confirmed in the NOESY spectrum of DXP-CB[8] (Supplementary Figure 34). That is because the peak of the alkene moiety Proton 2 shifted upfield and overlapped with those of the pyridine ring Proton 1 and 3 when DXP complexed with CB[8] (Supplementary Figure 32). Similar to DXBTZ, the upfield shift of all peaks assigned to the pyridine ring protons (1 and 3) and the aliphatic moiety protons (7 and 11) offered the insights into the J-aggregation interaction mode between DXP molecules in CB[8] (Supplementary Figure 32). The detailed discussion has been added on Page 11-12 of the revised manuscript. These NMR results, combined with the ultraviolet spectral redshift in Figure 2 as well as the results of control experiments in Figure 3 fully demonstrate the interaction mode.

(2) The ^1H NMR titration of host and guest is one of the best methods in studying the host-guest interaction in the absence of crystal structure. However, this paper did not analyze them in detail, and only described the improvement of water solubility. Obviously, the addition of the melon ring does not only increase the water solubility but also the shift of the proton signal, so the proper assignment of the proton signal facilitates the demonstration of the interaction modes in Figure 1.

Our Response: The signal assignment of the protons associated with the host-guest interaction has been conducted based on the newly-acquired 2D DQF-COSY and HSQC NMR data (Supplementary Figure 30-31 and Supplementary Figure 36-37), and the assignment results are shown in Supplementary Figure 32 (for DXP) and Supplementary Figure 38 (for DXBTZ). The chemical-shift due to the addition of the melon ring (CB[8]) has been analyzed and discussed in detail on Page 11-12 of the revised manuscript.

(3) The authors stated that the assembly of DXBTZ-CB[8] and CSA does not affect the extinction coefficient of DXBTZ-CB[8]. However, no UV and fluorescence spectral changes were found, what is the evidence for their assembly? Can TEM alone prove that these assemblies are highly stable and do not dissociate in the organism?

Our Response: The assembly of DXBTZ-CB[8] and CSA does not affect the extinction coefficient of DXBTZ-CB[8], because DXBTZ-CB[8]/CSA did have a similar extinction coefficient in the wavelength range of 500-900 nm, optoacoustic intensity at 692nm and fluorescence intensity at 820 nm to DXBTZ-CB[8] (Fig. 4c-4e). However, since the turbidity of DXBTZ-CB[8] changed slightly after the addition of CSA, the absorbance of generated DXBTZ-CB[8]/CSA in the wavelength range of 300-500 nm and 900-950 nm was quite different from that of DXBTZ-CB[8], thus providing some insights into their assembly (Fig 4c and its enlargement below). More importantly, the morphology, size, and surface charge of DXBTZ-CB[8] changed dramatically after adding CSA, which offered the intuitive evidence for the assembly of DXBTZ-CB[8] with CSA (Fig. 4a and 4b).

Enlargement of Figure 4c

As for the stability of DXBTZ-CB[8]/CSA, we have tested the its size change in H₂O, H₂O/DMEM and H₂O/FBS at different times. As shown in Supplementary Figure 42, the nanoagent DXBTZ-CB[8]/CSA is quite stable in pure water for 7 days, but slowly dissociates after 3 days in water containing DMEM medium or FBS, which was sufficient for targeted imaging and facilitated subsequent metabolic clearance from the body.

(4) What is the driving force of the assembly, electrostatic interactions or hydrogen bonding?

Our Response: It has been widely reported that CB[8]-based host-guest complex arise from a combination of multiple driving forces including ion-dipole, hydrogen-bonding, electrostatic, hydrophobic and π - π stacking interactions between macrocyclic CB[8] host molecules and guest molecules, thus typically exhibiting a high binding constant (*Angew. Chem. Int. Ed.*, 2021, **60**, 15166–15191; *Chem. Asian J.*, 2013, **8**, 1626-1632; *Chem. Rev.*, 2015, **115**, 7196-7239; *Chem. Soc. Rev.*, 2013, **42**, 9480-9508; *Small*, 2018, **14**, 1802234; *Chem. Rev.*, 2022, **122**, 9032-9077). As for the assembly of CSA with DXBTZ-CB[8], the driving force is manly the combination of hydrogen bonding, electrostatic and Van der Waals interactions, which is similar to the reported CSA-based nanoparticles (*Food Chem.*, 2022, **379**, 132163; *Adv. Healthcare Mater.*, 2021, **10**, 2100109; *Int. J. Pharm.*, 2016, **509**, 188-196; *Acta Chim. Sinica*, 2010, **68**, 1210).

(5) Fig 4g and 4j prove that DXBTZ-CB[8]/CSA only enters the cell through highly expressed CD44 receptors on the cell surface, which is insufficient. In Fig 4g, there is no red channel in HEK293 cells, indicating that DXBTZ-CB[8]/CSA cannot enter the cell, but the reasons why the material fails to enter the cell may be diverse. Other cells that do not express CD44 should be supplemented as a reference comparison. In Fig 4j, HA is used as a competitive inhibitor, but in addition to CD44, HA may also bind to other proteins on the cell surface to block DXBTZ-CB[8]/CSA from entering the cell.

Our Response: First, we would like to clarify there was a very weak red fluorescence signal and faint optoacoustic signal in CD44-low but not absent HEK293 cells, indicating that only a small amount of DXBTZ-CB[8]/CSA entered the cells (Fig. 4g and 4h). Since we are unable to obtain cells which absolutely do not express the CD44 receptor, we selected another four cell lines including CD44 low-expressing cells (L929 and L-02) and CD44 high-expressing cells (4T1 and T24) to validate that higher CD44 expression facilitates the uptake of CD44-targeted DXBTZ-CB[8]/CSA nanoagents into cells. From Fig. 4g and 4h, it was clear that, upon incubation with DXBTZ-CB[8]/CSA, significant OA signals and fluorescence could be observed in 4T1, T24 and MDA-MB-231 cells, whereas those were quite weak in the normal HEK293, L929 and L-02 cells. To further verify the CD44 targeting effect of DXBTZ-CB[8]/CSA, another CD44 inhibition experiment has been performed by using a specific inhibitor anti-CD44 antibody. As can be seen in Fig. 4i-j, the MDA-MB-231 cells pretreated with excess free HA or anti-CD44 antibody exhibited markedly lower optoacoustic signal and fluorescence than the untreated ones,

which confirmed that the endocytosis pathway of DXBTZ-CB[8]/CSA was mainly through CD44 receptor-mediated internalization.

(6) In Fig 5h and Fig 6b, DXBTZ-CB[8]/CSA enters the lesion site through in situ injection. Why did the photoacoustic imaging of DXBTZ-CB[8]/CSA disappear in the blank control group after just an hour? Is it metabolism? In Fig 5b, the material does not attenuate until 12 hours after entering the body.

Our Response: In Fig. 5h, the control mice or mice with orthotopic bladder tumor were intravesically injected with DXBTZ-CB[8]/CSA dispersion. Upon 1 h post administration, the dispersion was drawn back into the syringe, the catheter was withdrawn, and then the mice were imaged in the prone position. This procedure is equivalent to incubation of bladder tissue with DXBTZ-CB[8]/CSA. Therefore, as shown in Fig. 5h-5i, the normal bladder with low expression of CD44 only took up a small amount of nanoagents and exhibited a weak MSOT signal. On the contrary, the bladder inoculated with T24 cells which highly expressed CD44 receptors, absorbed a large amount of nanoagents and thereby displaying a strong MSOT signal.

In Fig. 6b, the control mice or mice with lymphatic metastasis was given DXBTZ-CB[8]/CSA dispersion via intra-footpad injection, but almost no MSOT signal could be observed at the popliteal lymph node site of control mice. That was because there were no cells with high expression of CD44 receptors in the lymphatic system of control mice, and the administrated nanoagents could not be enriched in large quantities at the lymph node (imaging site). In contrast, for the modeling mice, their lymphatic system (including the popliteal lymph node and lymphatic vessels) had been invaded by the CD44-overexpressing 4T1 cells which spread from the primary tumor site (footpad). After injection into the footpad, the administrated nanoagents could be strongly taken up by the metastatic 4T1 cells present in the lymphatic system, and thus strong MSOT signal could be visualized at the popliteal lymph node site of mice with lymphatic metastasis as shown in Fig. 6b-6c and 6e.

The MSOT intensity difference in Fig 5h and Fig 6b, was attributed to the difference in the amount of absorbed contrast agents resulting from the difference in the amount of CD44 receptors at the imaging site between the blank control group and the modeling group, rather than the metabolic process. Dosing and imaging procedures for all mouse models are described in detail in the *Methods* Section on Page 32 in the revised manuscript.

(7) In Fig 7d, DXBTZ-CB[8]/CSA is heavily enriched in the AKI model (injection concentration of 62 mg/kg). At the same time, in Fig 5f, this material is also widely enriched in the kidneys (injection concentration of 46.7 mg/kg) and tumor sites. Will the difference between the data in Fig 7d and Fig 5f be caused by the different concentration of injection?

Our Response: The mean MSOT intensity ($\approx 229 \pm 37$ a.u.) in kidneys of the mice with subcutaneous tumor (Fig. 5f, injection concentration of 46.7 mg/kg) was lower than that (\approx

260 ± 130 a.u.) in kidneys of the control mice (Fig. 7d, injection concentration of 62.2 mg/kg), which could be roughly considered to be caused by different doses of the injections, even though these two groups are not comparable.

The imaging results of AKI mice should be compared to those of control normal mice (without any tumor or lesion). It can be clearly seen from Fig. 7c-7d, that strong MSOT signal ($\approx 1270 \pm 270$ a.u.) of DXBTZ-CB[8] could be observed in the kidneys of the AKI mouse, while the control normal mouse only exhibited a weak MSOT signal ($\approx 260 \pm 130$ a.u) in the kidney region. Compared with control mice, expression of CD44 receptors is significantly upregulated on the tubuloepithelial cells and the permeability of inflamed renal microvasculature is enhanced in AKI mice. Therefore, the difference in MSOT intensity between the control normal mice and AKI mice at the same injection concentration was attributed to the difference in the amount of nanoagents accumulated in the kidney.

Other minor questions:

(1) The diffusion coefficient is an important data for supramolecular assemblers (DXBTZ-CB[8] and DXP-CB[8])

Our Response: 2D DOSY experiments has been performed to calculate the diffusion coefficient, and the results are shown in Supplementary Figure 35 and Supplementary Figure 41. The diffusion coefficient of DXP-CB[8] and DXBTZ-CB[8] was determined as $1.7115 \times 10^{-10} \text{ m}^2 \cdot \text{s}^{-1}$ and $1.5115 \times 10^{-10} \text{ m}^2 \cdot \text{s}^{-1}$, respectively (Page 12 of the revised manuscript).

(2) The content of Figure 2e is inconsistent with the description in the text.

Our Response: We thank the reviewer for pointing out this typo. The statement “..., but the fluorescent emission at 760 nm decreased remarkably (Fig. 2e)” has been corrected to “..., but the fluorescent emission at 760 nm decreased remarkably (Fig. 2d)” on Page 7 of the revised manuscript.

(3) How does the assemblies metabolic clearance from the tumor?

Our Response: The clearance behavior of the nanoagents has been measured, and the results are shown in Supplementary Figure 49. Approximately 90% of the injected nanoagents could be excreted into feces and urine through metabolism within 7 days after administration. The relevant discussion has been added on Page 18 of the revised manuscript.

Reviewer #2 (Remarks to the Author):

Multispectral optoacoustic imaging is an interesting approach to promote the spatial resolution and tissue penetration depth and has a major impact on the field of preclinical and clinical applications. In this manuscript, the authors reported two water-soluble supramolecular assemblies based on cucurbit[8]uril and validated their utility in several animal models. The constructed CB[8]-based assemblies exhibited significantly enhanced optoacoustic performance compared to the individual guest organic small-molecule chromophore. After being coated with targeting components, the assemblies successfully achieved the diagnosis and localization of subcutaneous tumor, orthotopic bladder tumor, lymphatic metastasis of tumor and renal ischemia/reperfusion injury in mice by multispectral optoacoustic imaging. Overall, this work is rather interesting and is well presented with proper discussions.

Our Response: We thank the reviewer for the useful comments and advice. We have revised the manuscript accordingly.

1. Statistical analysis of the data should be provided, and this should be added especially for the data in Fig. 5-7.

Our Response: We thank the reviewer for the comments and suggestions. We have performed statistical analysis on all the data. Significance test results for biologically relevant data have also been added in Figure 5-7.

2. The source of cryosection images shown in Fig. 5j, 6d, 7e and Supplementary Fig. 36b for reference to anatomical details should be stated in the Methods Section.

Our Response: We have added the statement “Cryosection images provided with *viewMSOT* software in the MSOT system are used for reference to anatomical details” in the *Methods* Section on Page 32 of the revised manuscript.

3. In Supplementary Fig. 23, the optoacoustic spectrum (a or b) should be clearly assigned to DXP (or DXBTZ)-CB[8] assemblies.

Our Response: We have clarified the optoacoustic spectrum (a or b) in assigned to DXP (or DXBTZ)-CB[8] assemblies in the caption of Supplementary Fig. 23.

4. In Supplementary Fig. 34, the format of horizontal axis labels should be optimized for readability.

Our Response: We have redrawn Supplementary Fig. 47 (original Supplementary Fig. 34) in the revised Supplementary Information. The format of horizontal axis labels has also been optimized so that the content in the graph is now clear and easy to read.

Reviewer #3 (Remarks to the Author):

In this manuscript, the supramolecular complexes based on the host-guest chemistry of cucurbit[8]uril and dioxanthene derivative were developed for multispectral optoacoustic imaging of tumors and so forth. The selling point of this work should result from water solubility of assemblies, that is, the DXBTZ-CB[8]. Unfortunately, the assemblies and nanoparticles with CD44-targeting capability are dispersed solid systems, which compromises the novelty and significance of this present work. Also, the introduction of targeting molecules (CSA) on the drug nanoparticles is not a new concept. On the whole, the approach and material property delivered in this work have been broadly reported. It can only be considered an extension of the cucurbit[8]uril host-guest chemistry for preparation of dioxanthene derivative contrast agents for imaging application. Therefore, rejection should be recommended and the following questions are suggested consideration for submission elsewhere.

Our Response: We thank the reviewer for the useful comments and advice. We have revised the manuscript accordingly. We sincerely hope that the revised manuscript is now suitable for publication.

Major points

- Cucurbit[8]uril has been broadly used as the building block for host-guest supramolecular chemistry. The incorporated contrast agents into CB[8] can be easily imagined. Therefore, it is the overstatement of the strategy reported in this work regarded as the newly-developed easy-to-follow approach.

Our Response: We thank the reviewer for the comment, which along with his/her raised questions would help us improve the overall quality of the manuscript. However, we would like to clarify the significance of our work first. As articulated in our manuscript, the biggest selling point of this work is that we report for the first time the use of cucurbit[8]uril to improve the optoacoustic performance of guest chromophore molecules, and explain its mechanism (by simultaneously increasing the extinction coefficient and decreasing the fluorescence emission of guest chromophore molecules) in detail (Fig. 2). At the same time, the water dispersibility of guest chromophore is also greatly ameliorated after complexing with CB[8], which is conducive to its application in biological fields. As for CSA, we only use it as an example with targeting function to demonstrate the potential of the formulated host-guest complexes with enhanced optoacoustic performance for application in biomedical imaging.

Different from the previously reported methods (*Chem. Soc. Rev.*, 2021, **50**, 7924-7940; *Nat. Methods*, 2016, **13**, 639-650; *Chem. Soc. Rev.*, 2019, **48**, 2053-2108; etc.), which often involve complicated chemical synthesis, this approach based on host-guest supramolecular chemistry is easy-to-follow, and can solve the problems of relatively bad optoacoustic performance and poor water dispersibility of organic small-molecule chromophores at one time.

- Multispectral optoacoustic imaging technique shows the advantageous in comparison to the other optical imaging ones, but it is unclear how the imaging agents impede or facilitate its applications preclinically or clinically. In other words, how can the development of photoacoustic contrast agents fasten the clinical applications of multispectral optoacoustic imaging technique?

Our Response: Besides the advantages of traditional optoacoustic imaging, multispectral optoacoustic imaging also possesses the unique ability to discriminate specific chromophore (contrast agent) from background absorbers (e.g. hemoglobin) in tissue based on the differences in their absorption spectral signatures. The high performance of a multispectral optoacoustic imaging depends to a large extent on the interplay of the instrumentation/algorithm of the imaging system and the contrast agent used (*Nature Photon.*, 2015, 9, 219–227; *Chem. Rev.*, 2010, 110, 2783–2794; *Nat. Methods*, 2016, 13, 627–638). Generally speaking, the contrast agents with following features (1. a characteristic absorption spectrum with significant steep peak(s) in the near-infrared range; 2. a high extinction coefficient for maximizing light absorption; 3. high photostability after light irradiation; 4. high heat conversion efficiency for maximizing the conversion of light to heat and generation of strong acoustic signals; and 5. low quantum yields of competitive deactivation pathways of excited state (e.g. fluorescence, phosphorescence, and energy transfer) for minimizing the energy losses.) are best suited for multispectral optoacoustic imaging. The more obvious the above-mentioned characteristics of the contrast agent are, the more favorable it is for the spectral unmixing process in multispectral optoacoustic imaging, thereby improving the imaging sensitivity and realizing wider applications. (*Chem. Soc. Rev.*, 2021, 50, 7924–7940; *Acc. Chem. Res.*, 2018, 51, 2897–2905; *Theranostics*, 2019, 9, 1550–1571)

In addition to those requirements on photophysics, the preferable contrast agents for clinical applications should also exhibit the following physical, chemical and biological properties (1. good water solubility or dispersibility; 2. high chemical stability; 3. excellent biocompatibility; 4. low toxicity and immunogenicity for safe use; 5. passive or active targeting ability for improving accumulation at tissues of interest). Therefore, the development of optoacoustic contrast agents with superior performance to meet the above requirements is necessary to fasten the clinical applications of multispectral optoacoustic imaging technique.

- The authors stated “physical encapsulation often leads to broadened as well as decreased absorption peak(s) and subsequent reduction in optoacoustic signal due to the random aggregation of contrast agents.” The reviewer can not follow the translation of reduction in acoustic signal as the random aggregation of contrast agents. One can well understand the reduction of fluorescent signal is due to the aggregation induced quenching, but not for photoacoustic agents. They share different the mechanism with that the excited state down to the ground state in different pathway. The thermal relaxation is not subjected to the

aggregation of molecular chromophores, oppositely, the aggregation may enhance the acoustic signal due to the blocking of other energy pathways.

Our Response: Optoacoustic effect relies on the efficiency of converting absorbed optical energy into thermal energy and then into detectable acoustic waves, therefore the optoacoustic performance of a chromophore is highly dependent on its absorption coefficient and deactivation pathways of its excited state (*Science*, 2012, **335**, 1458-1462; *Angew. Chem. Int. Ed.*, 2020, **59**, 11717-11731). As shown in Fig. 4c, in the absence of CB[8], the mixture of DXBTZ and CSA (DXBTZ/CSA) exhibited a significantly blue-shifted and broadened absorption peak with decreased extinction coefficient, which was caused by the random aggregation. Given that the relatively low fluorescence quantum yield of NIR chromophores, reduction in optoacoustic signal of DXBTZ after directly mixing with CSA (the physical encapsulation process) was mainly attributed the decreased extinction coefficient in near-infrared range (Fig. 4d). Therefore, we stated that “physical encapsulation often leads to broadened as well as decreased absorption peak(s) and subsequent reduction in optoacoustic signal due to the random aggregation of contrast agents”. Such a phenomenon was also found in previously reported work (*Adv. Funct. Mater.*, 2019, **29**, 1807960).

- No experimental methods were showed on how the nanorods were prepared and corresponding conditions. It is unclear why the resulting assemblies are the rods, but not other structures. No experimental data and discussion on the storing stability that are of key importance for supramolecular nanodrugs. Do the nanorods evolve over aging time, e.g. months or longer? What maintains the stability of nanorods?

Our Response: The preparation method and storage conditions of binary host-guest complexes have been described in detail in the **Methods Section on Page 31** of the revised manuscript. Similar to previously reported CB[8]-based binary complexes, inclusion of dixanthene derivatives into CB[8] following the sled binding mode would result in linear host-guest assembly units, and many assembly units could further stack with each other for the final nanorod-like morphology (*Angew. Chem. Int. Ed.*, 2018, **57**, 12519-12523; *Mater. Today Chem.*, 2022, **25**, 100954; *J. Photochem. Photobiol. A: Chem.*, 2018, **355**, 419-424; *Chem. Comm.*, 2019, **55**, 10654-10664; *Angew. Chem. Int. Ed.*, 2016, **55**, 11452-11456). The formation of stacked nanorods is shown below:

Scheme

As for the storing stability, we have measured the size of DXP-CB[8] and DXBTZ-CB[8] in aqueous solution for different time and the result was shown in Supplementary Figure 24. Apparently, both complexes were stable for several weeks at room temperature. Each binary complex unit is positively charged, so the stability of nanorods is maintained by the balance between electrostatic repulsion and van der Waals attraction.

- It is inappropriate to state the assemblies can be water soluble as either nanorods and nanoparticles are clearly dispersed solid system, same to all the nanodrugs dispersed in water reported previously. It is suggested to clearly distinguish the solid dispersion system from water soluble system. They are completely different phases.

Our Response: We thank the reviewer for this suggestion. We have corrected the inappropriate term “water-soluble or water solubility” to “water-dispersible and water dispersibility” throughout the revised manuscript.

- It is confusing to follow the morphological transformation from nanorods to nanoparticles after mixing with CSA. How to ensure the stable host-guest complexation between dixanthene derivative and CB[8] in the presence of CSA? The dixanthene derivative and CB[8] assemblies should be nanorods, looking like a thermodynamically favorable crystal. In principle, the coating can not change the inherent nanorod morphology of binary complexes. How does CSA alter the morphology of resulting assemblies? The morphological changes hint the multiple interactions of CSA, dixanthene derivative and CB[8]. What is the mechanism?

Our Response: The binding of CB[8] to guest molecules is a strong interaction. In our manuscript, the association constants (K_s) for complexation of CB[8] with DXBTZ have been determined as $5.84 \times 10^8 \text{ M}^{-1}$, indicating the strong host-guest interaction. Therefore, the addition of negatively-charged CSA only break the stacking of the binary complex units, but not destroy the inclusion between DXBTZ and CB[8], as illustrated in the Scheme above. In addition, the extinction coefficient in the wavelength range of 500-900 nm, optoacoustic intensity at 692nm and fluorescence intensity at 820 nm of DXBTZ-CB[8]/CSA is similar to DXBTZ-CB[8], which fully prove the point (Figure 4c-4e).

The formation of DXBTZ-CB[8]'s nanorod morphology was the result of stacking of binary complex units. However, the stacking is a weak balance of electrostatic repulsion and van der Waals attraction. Therefore, the addition of negatively-charged CSA can break the stacking of the binary assembly unit and coat on each binary complex unit through hydrogen bonding, electrostatic and Van der Waals interactions to form the spherical morphology, as displayed in the Scheme above (*Food Chem.*, 2022, **379**, 132163; *Adv. Healthcare Mater.*, 2021, **10**, 2100109; *Int. J. Pharm.*, 2016, **509**, 188-196; *Acta Chim. Sinica*, 2010, **68**, 1210).

- In the tumor imaging sections, one can not follow how well the nanoparticles of DXBTZ-CB[8] complexes for imaging the tumors as the positive controls with other contrast agents such as

gold nanoparticles are missing.

Our Response: We have added the positive control experiments by using the commercially available PEG-functionalized gold nanorods (AuNR-PEG) as the contrast agent, and then compare its optoacoustic performance with that of our prepared nanoagent DXBTZ-CB[8]/CSA at the same mass concentration. As shown in Supplementary Figure 48, although these two contrast agents have similar absorption and optoacoustic spectra in the wavelength range of 680-900 nm, our prepared DXBTZ-CB[8]/CSA nanoagents displayed superior photostability, and generated higher optoacoustic signal output than AuNR-PEG both in vitro and in vivo. The results imply that the nanoagent DXBTZ-CB[8]/CSA can exhibit higher signal-to-noise ratio in multispectral optoacoustic imaging and is more suitable for long-term tracking and monitoring. The relevant discussion has been added to page 17 of the revised manuscript.

Minor points

- How to experimentally determine all the resulting complexes being binary, but not mixture of various complexes? The direct evidence of binary complexes is highly appreciated.

Our Response: The most direct evidence for binary complexes is the crystal structure. But unfortunately, as with most reported CB[8]-based complexes, we did not obtain usable crystals of dioxanthene derivative-CB[8] complexes after three month of trials. Even so, the data in the revised manuscript can also illustrate that the resulting dioxanthene derivative-CB[8] complexes are binary.

First, the Job's plots in Supplementary Figure 25 demonstrated that dioxanthene derivative-CB[8] complexes ostensibly adopted a stoichiometric ratio of 1:1. Second, the association constants (K_s) for complexation of CB[8] with DXP or DXBTZ have been determined as $4.36 \times 10^8 \text{ M}^{-1}$ and $5.84 \times 10^8 \text{ M}^{-1}$ respectively, indicating the strong host-guest interaction between dioxanthene derivatives and CB[8] (Supplementary Figure 26). Therefore, equimolar amount of dioxanthene derivatives and CB[8] in aqueous environment produced only the binary complexes instead of mixture of various complexes. To further prove this point, a lot of 1H NMR and 2D NMR (including NOESY, DQF-COSY, HSQC and DOSY) have been performed for DXP-CB[8] and DXBTZ-CB[8], and the results are shown in Supplementary Figure 28-41. The detailed discussion has been added on Page 11-12 of the revised manuscript.

- How was polysaccharide chondroitin sulfate A (CSA) coated on DXBTZ-CB[8]?

Our Response: The host-guest complexation between DXBTZ and CB[8] has a high binding constant, so the addition of CSA had little effect on the binary complex units. The interaction between each binary complex unit is relatively weak, so the negatively-charged CSA can break the stacking of the binary assembly unit and coat on each binary complex unit through hydrogen bonding, electrostatic and Van der Waals interactions, as displayed in the Scheme above (*Food Chem.*, 2022, **379**, 132163; *Adv. Healthcare Mater.*, 2021, **10**, 2100109; *Int. J. Pharm.*, 2016, **509**, 188-196; *Acta Chim. Sinica*, 2010, **68**, 1210). The relevant discussion has

been reflected Page 14 of the revised manuscript.

- What is the solubility of the two positively charged chromophores, dixanthene derivatives (DXP and DXBTZ) in water?

Our Response: The solubility of DXP in water was determined to be 1.06 mM/L. However, the amount of DXBTE dissolved in pure water is too small to be weighed precisely. Therefore, the solubility of DXBTZ in water was only determined to be < 0.06 mM/L.

- The absorption less than 700 nm can not be considered near-infrared absorption. Red light absorption should be more accurate.

Our Response: We agree that the strict definition for near-infrared light is above 700 nm. However, many literatures have demonstrated that the range of NIR was within 650-900 nm. (*Angew. Chem. Int. Ed.*, 2022, **61**, e202209793; *J. Am. Chem. Soc.*, 2017, **139**, 13243; *ChemistryOpen*, 2019, **8**, 1407-1409)

- In Supplementary Figure 29, the stability of nanoparticles was only evaluated for 24h at most. This is insufficient and the extended aging time is encouraged.

Our Response: We have extended the aging time to evaluated the stability of nanoagents, and the result was shown in **Supplementary Figure 42**. The nanoagent DXBTZ-CB[8]/CSA was quite stable in pure water for 7 days, but slowly dissociated after 3 days in water containing DMEM medium or FBS. These results indicated the properties of DXBTZ-CB[8]/CSA hold promise for targeted imaging and subsequent metabolic clearance in biological applications. The relevant discussion has been added on **page 14** of the revised manuscript.

- In Figure 4c, the complete overlapping of red and green curves is questionable. This should be double checked. The turbidity in the two cases should be quite different. Thus, it is impossible to have two completely overlapped absorption curves.

Our Response: We have double checked this figure, and confirmed it is okay. In addition, the data points of these lines have been given in the **Source Data** profile. Actually, these two lines representing the absorption of DXBTZ-CB[8] and DXBTZ-CB[8]/CSA do not exactly overlap. They just have similar extinction coefficient in the wavelength range of 500-900 nm, indicating that the assembly between DXBTZ-CB[8] and CSA only break the stacking of the binary assembly units, but not destroy the inclusion between DXBTZ and CB[8]. In contrast, the absorption of DXBTZ-CB[8] in the wavelength range of 500-900 nm changed significantly after the addition of CSA, as the solution turbidity slightly changed. A magnified view including these two lines is also shown below:

Enlargement of Figure 4c

- The details on the measurement of flow cytometry for verifying the uptake of nanoparticles should be supplemented. What is responsible for the fluorescence of cells after uptake of nanoparticles? Does this mean the decomplexation of binary complexes and release of molecular contrast agents?

Our Response: The details on the measurement of flow cytometry for verifying the uptake of nanoparticles have been supplemented in *Supplementary Methods* Section on Page S8-S9 of the revised Supplementary Information. The increase in cellular fluorescence intensity means the increase in cellular uptake of the nanoagents, rather than the decomplexation of binary complexes and release of molecular contrast agents. To further prove this point, we measured the changes in optical density at 692 nm of cells incubated with the nanoagent for different times, and the result was displayed in Supplementary Figure 44. It was obvious that the optical density at 692 nm of cells also increased with the prolongation of incubation time, fully supporting the above notion.

Reviewers' Comments:

Reviewer #1:

Remarks to the Author:

Since the authors have clarified almost all the issues concerned, I suggest the acceptance of this paper for publication in this journal in the present form.

Reviewer #2:

Remarks to the Author:

In this revised version, the authors fully addressed my earlier comments, thus the manuscript is suitable for publication.

Reviewer #3:

Remarks to the Author:

I carefully looked through the response letter and manuscript. The revised manuscript improved the specific technical quality, but not delivered new concept and/or newly-developed system for biomedical application. Thereby, it is best suited to a topically-focused journal after considering the following questions.

1. "host-guest supramolecular chemistry is easy-to-follow". this is true compared to the organic synthesis, but this is a general recognition in the scientific community. What is the unique advance of this work in methodology?
2. The introduction of CSA resulted in the structural transformation and different delivery effect in vivo, but the authors fail to clearly show the mechanism of structural transformation and its advantages/disadvantages. Does it mean the supramolecular structures could alter if the coating agents were changed? It is unclear.
3. No direct evidence (such as single crystal XRD) of binary complexes was provided, which compromised the reliability of the conclusions.
4. The method for complexation of CB[8] with DXBTZ has been broadly reported. Therefore, it is suggested that the manuscript more focused on the photoacoustic imaging application.
5. From the point of view of photophysics, the absorption less than 700 nm can not be considered as the near-infrared. This reviewer would like to suggest "deep red light" absorption, but not following the inaccurate description as referenced.

Response to Reviewers' Comments

Reviewer #1 (Remarks to the Author):

Since the authors have clarified almost all the issues concerned, I suggest the acceptance of this paper for publication in this journal in the present form.

Our Response: We thank the reviewer for the recommendation of publication.

Reviewer #2 (Remarks to the Author):

In this revised version, the authors fully addressed my earlier comments, thus the manuscript is suitable for publication.

Our Response: We thank the reviewer for the recommendation of publication.

Reviewer #3 (Remarks to the Author):

I carefully looked through the response letter and manuscript. The revised manuscript improved the specific technical quality, but not delivered new concept and/or newly-developed system for biomedical application. Thereby, it is best suited to a topically-focused journal after considering the following questions.

Our Response: We thank the reviewer for the comments. We have revised the manuscript accordingly. We sincerely hope that the revised manuscript is now suitable for publication.

1. "host-guest supramolecular chemistry is easy-to-follow". this is true compared to the organic synthesis, but this is a general recognition in the scientific community. What is the unique advance of this work in methodology?

Our Response: We thank the reviewer for the comment. We would like to clarify that our work mainly focuses on providing an easy-to-follow strategy/method to improve the optoacoustic performance of optoacoustic contrast agents. From the perspective of developing excellent optoacoustic contrast agents, we report that the optoacoustic performance of contrast agents could be greatly enhanced by simple complexation with CB[8]. This easy-to-follow method based on supramolecular chemistry to enhance the properties of optoacoustic contrast agents has not been reported yet and can circumvent the difficulties brought about by conventional complex organic synthesis routes, which is the unique advance for the approach of this work in the field of optoacoustic imaging. From the supramolecular chemistry point of view, although our CB[8]-based systems share the common preparation method with the widely reported ones, the developed assemblies realize the regulation of the optoacoustic performance of guest contrast agents, which has never been investigated preciously. Thus, the work in our manuscript pioneered the application of CB[8] in this regard. Overall, one on hand, our work proposes an easy-to-follow approach/strategy based on supramolecular chemistry to modulate the optoacoustic performance of contrast agents in the field of optoacoustic imaging; on the other hand, this work expands the application field of CB[8]-based supramolecular chemistry. These pertinent narratives have been included in the *Introduction and Conclusion section*.

2. The introduction of CSA resulted in the structural transformation and different delivery effect in vivo, but the authors fail to clearly show the mechanism of structural transformation and its advantages/disadvantages. Does it mean the supramolecular structures could alter if the coating agents were changed? It is unclear.

Our Response: As shown in Fig. 1 below, the mechanism of structural transformation is that CSA can break the stacking (i.e., nanorods) of the binary assembly unit (not destroy the inclusion between guest molecule and CB[8]), and *encapsulate* each binary complex unit to form new nanostructures (spherical morphology) through hydrogen bonding, electrostatic and

van der Waals interactions. The relevant discussion is on Page 14 in the revised manuscript. Such structural transformation of host-guest complexes induced by coating agents is a common phenomenon in supramolecular chemistry, which has been broadly reported (e.g., *Angew. Chem. Int. Ed.* 2022, **61**, e2022067; *ACS Appl. Mater. Interfaces* 2022, **14**, 4417-4422; *Angew. Chem. Int. Ed.* 2021, **60**, 3870-3880; *Chem. Commun.* 2020, **56**, 10113-10126; *J. Am. Chem. Soc.* 2019, **141**, 6583-6591; *Chem. Commun.* 2019, **55**, 4343-4346; *Angew. Chem. Int. Ed.* 2018, **57**, 12519-12523). Due to the limited number of references, we cited a well-explained paper (Ref. 39: *Angew. Chem. Int. Ed.* 57, 12519–12523 (2018)) related to our discussion on Page 14 in our manuscript.

Fig. 1 Schematic illustration for the formation of DXBTZ-CB[8]/CSA assemblies.

In addition, the introduction of CSA can shield the positive charges of host-guest complexes and reduce their size distribution to avoid rapid clearance from the blood system. More importantly, the CSA coating can maintain the enhanced optoacoustic performance generated by host-guest interaction and endow the resulted assemblies with unique functionality for better applications. These advantages have been included in the discussion on Page 13-14 in our manuscript. As for the adverse effects of CSA coating on optoacoustic performance and applications, we haven't found them yet.

To demonstrate that the coating agent can alter the morphology of supramolecular structures, we performed a TEM experiment on DXBTZ-CB[8] complexes after addition of another negatively-charged polysaccharide macromolecule, hyaluronic acid (HA). As displayed in the TEM image (Fig. 2) below, HA coating also changed the DXBTZ-CB[8] nanorod morphology into spherical morphology, which is similar to the results of coating CSA onto DXBTZ-CB[8].

Fig. 2 A typical TEM image for HA coating onto DXBTZ-CB[8] complexes.

3. No direct evidence (such as single crystal XRD) of binary complexes was provided, which compromised the reliability of the conclusions.

Our Response: As we stated in the previous response letter, usable single crystals of dioxanthene derivative-CB[8] complexes for X-ray diffraction could not be obtained after many attempts. The possible reason is that the J-aggregates formed by the sled n:n binding mode between CB[8] and dioxanthene derivatives are similar to polymers (well proven by 2D DOSY results in Supplementary Figure 35 and 40), which are not conducive to the formation of single crystals. In the cases without a single crystal, the ^1H NMR titration and 2D NMR experiments are often used to provide direct evidence for the study of host-guest complexes, because the interactions between the protons assigned to host/guest molecules can be clearly deduced from the results (e.g., *Nat Commun.* 2023, **14**, 518; *Nat. Mater.* 2022, **21**, 103-109; *Nat. Chem.* 2022, **12**, 808-813; *Nat. Commun.* 2022, **13**, 7046; *Nat. Commun.* 2022, **11**, 4655; *J. Am. Chem. Soc.* 2022, **144**, 6483-6492; *Angew. Chem. Int. Ed.* 2022, **134**, e202115265; *Angew. Chem. Int. Ed.* 2021, **133**, 6691-6697; *Angew. Chem. Int. Ed.* 2020, **59**, 18748-18754; *Angew. Chem. Int. Ed.* 2019, **58**, 10553-10557; *J. Am. Chem. Soc.* 2017, **139**, 3202-3208; *Nat. Commun.* 2016, **7**, 1158; *J. Am. Chem. Soc.* 2014, **136**, 6602-6607; *Nat. Chem.* 2013, **5**, 376-382; *Science* 2012, **335**, 690-694).

In the ^1H NMR spectra of dioxanthene derivatives with varied amounts of cucurbit[8]uril, clear and regular shifts of the proton peaks were observed, confirming the complexation between cucurbit[8]urils and dioxanthene derivatives (Supplementary Figure 28 and 29). For DXP and DXP-CB[8], the assignment of proton peaks was analyzed based on DQF-COSY and HSQC (Supplementary Figure 30 and 31). As shown in Supplementary Figure 32, the peaks at $\delta = 8.4$ and 7.5 are assigned to the protons of pyridine rings (1 and 3); the peaks at $\delta = 7.5$ and 6.1 are assigned to the protons of alkene moiety (2 and 6); and the peaks at $\delta = 4.3$ and 1.5 are assigned to the protons of aliphatic moiety (7 and 11). In general, the nuclear Overhauser

effect (NOE) occurs when the distance between two protons is less than 5 Å. Thus, 2D NOESY is often used to study the supramolecular stereo information. Within the DXP molecule, most of the distance between the aliphatic chain proton (7 or 11) and the pyridine ring proton (1 or 3) is already less than 5 Å (Supplementary Figure 33). In contrast, the distance between the aliphatic chain proton (7 or 11) and the alkene moiety proton (2) is much larger than 5 Å in a single DXP molecule, so that the appearance of NOE correlation peak between these protons could theoretically verify the interaction of DXP molecules in the CB[8] cavity. We found that the NMR signal of Proton 2 of DXP shifted upfield and overlapped with those of proton 1 and 3 when DXP complexed with CB[8]. For this reason, the NOESY spectrum of DXP-CB[8] did not show intuitive evidence for the interaction mode of DXP molecule in the CB[8] cavity (Supplementary Figure 34). Thus, the ¹H NMR spectra of DXP and DXP-CB[8] provided insights into the proposed binding mode. As shown in Supplementary Figure 32, all peaks of the pyridine ring protons (1 and 3) and the aliphatic moiety protons (7 and 11) underwent a pronounced upfield shift after DXP complexation with CB[8], caused by the shielding effect of another DXP molecule in the cavity, indicating the formation of J-aggregates. The shielding effect of the CB[8] cavity on the other side of the DXP molecule also contributes to the upfield shift, illustrating the interaction between CB[8] and DXP.

Similarly, the assignment of peaks was also analyzed based on DQF-COSY and HSQC (Supplementary Figure 36 and 37), and the results were shown in Supplementary Figure 38. Although the peaks assigned to aromatic protons of DXBTZ only exhibited broadened signals due to π - π stacking of their dioxanthene core in the solvent, the peak belonging to the hydrophilic methyl group proton (11) remained very clear. As seen in Supplementary Figure 39 and 40, the distance between the aromatic proton 4 and methyl group proton 11 was greater than 5 Å in a single DXBTZ molecule, while the cross peak between these two protons appeared in the NOESY spectrum of DXBTZ-CB[8], demonstrating that DXBTZ molecule could only form J-aggregates to reduce the spacing between proton 4 of one DXBTZ molecule and proton 11 of the other one in the CB[8] cavity. Moreover, the ¹H NMR changes of DXBTZ before and after complexation with CB[8] could also provide evidence for the proposed sled n:n binding mode. From Supplementary Figure 37, the broad NMR peaks of DXBTZ transformed into sharp and clear ones upon the addition of CB[8], suggesting the complexation between DXBTZ and CB[8]. At the same time, the peak assigned to proton 11 underwent an upfield shift, indicating that the methyl group was strongly shielded by the host CB[8] molecule and the benzothiazole ring of the other DXBTZ molecule in the CB[8] cavity. Therefore, it can be deduced that the guest molecule DXBTZ adopts a sled n:n interaction mode when complexing with the host molecule CB[8]. These NMR results were clearly discussed to demonstrate the formation of binary complexes on Page 11-12 in the manuscript.

Although we were unable to obtain the single crystal XRD data, we conducted powder X-ray diffraction experiments to support the above conclusions. As shown in Fig. 3 below, compared with the XRD patterns of DXP/DXBTZ or CB[8] alone, the XRD pattern of DXP-CB[8]/DXBTZ-CB[8]

displays some new peaks, implying the formation of binary complexes.

Fig. 3. Powder X-ray diffraction of powder samples (A) CB[8], DXP as well as DXP-CB[8] and (B) CB[8], DXBTZ as well as DXBTZ-CB[8]. The black dotted lines indicate new peaks appearing in the XRD pattern of DXP-CB[8] or DXBTZ-CB[8] complex powders.

4. The method for complexation of CB[8] with DXBTZ has been broadly reported. Therefore, it is suggested that the manuscript more focused on the photoacoustic imaging application.

Our Response: We acknowledge that complexation of CB[8] with the model guest molecules DXP and DXBTZ share the common preparation procedures with the reported ones. The research focus of this manuscript is to provide an easy-to-follow strategy/method to enhance the optoacoustic performance of guest molecules rather than just preparing some CB[8]-based assemblies. Since CB[8] has never been used to regulate the optoacoustic properties of guest molecules, as a proof of concept, we prepared two complexes by complexation of CB[8] with model guest molecules DXP and DXBTZ, and explained the mechanism of enhanced optoacoustic performance in detail (Figure 2-3 in the manuscript). Indeed, we gave more focus on applying the prepared systems for photoacoustic imaging applications in the manuscript. Our studies fully demonstrated the in vitro and in vivo applications of the prepared assemblies to verify their effectiveness on photoacoustic imaging (Figure 4-7 in the manuscript).

5. From the point of view of photophysics, the absorption less than 700 nm can not be considered as the near-infrared. This reviewer would like to suggest “deep red light” absorption, but not following the inaccurate description as referenced.

Our Response: Thanks for the reminder. We have replaced the term “near-infrared” with “deep red light” in the revised manuscript accordingly.

Reviewers' Comments:

Reviewer #3:

Remarks to the Author:

The reviewer could understand the efforts made by the authors in improving the quality of their work. I totally agree the importance of optoacoustic imaging of supramolecular nanoparticles for promoting the development of this technique in biomedical field. However, the mechanism for formation of supramolecular nanostructures is questionable, in particular the transformation of nanorods to nanoparticles. The presence of CSA does cause the complete alteration of assembled nanostructures, but the translation on the coating on the nanorods is incorrect. The structural change implies the strong interactions between DXBTZ-CB[8] and CSA. It may be more accurate to consider the formation of DXBTZ-CB[8]/CSA nanoparticles as a coassembly process. Following this, the mechanism for coassembly should be clearly discussed.

The statement on "easy-to-follow method based on supramolecular chemistry" should be weakened as many many reports including those presented in response letter by the authors have worked the supramolecular method and strategy. So the reviewer would like to suggest more emphasis on the performance of optoacoustic contrast agents, which have been well demonstrated in this work.

Overall, the concept delivered in this work is not attractive, but the display on the performance of optoacoustic imaging of assembled nanomaterials (NPs) is of interest. The comments can be a reference for the authors before further consideration for publication.

Response to Reviewers' Comments

Reviewer #3 (Remarks to the Author):

The reviewer could understand the efforts made by the authors in improving the quality of their work. I totally agree the importance of optoacoustic imaging of supramolecular nanoparticles for promoting the development of this technique in biomedical field. However, the mechanism for formation of supramolecular nanostructures is questionable, in particular the transformation of nanorods to nanoparticles. The presence of CSA does cause the complete alteration of assembled nanostructures, but the translation on the coating on the nanorods is incorrect. The structural change implies the strong interactions between DXBTZ-CB[8] and CSA. It may be more accurate to consider as a coassembly process. Following this, the mechanism for coassembly should be clearly discussed.

Our Response: We thank the reviewer for the useful comments and suggestions. To explain the formation mechanism of DXBTZ-CB[8]/CSA nanoagents, time-dependent TEM and DLS experiments were conducted to monitor the transformation process of nanorods to nanoparticles, and the results were shown in Supplementary Figure 42. As seen from Figure below (a combination of Fig. 2L, Supplementary Figure 42 and Fig. 4b), driven by the negative charges of CSA, the nanorods (Figure A and E) stacked by positively-charged binary unit DXBTZ-CB[8] rapidly dissociated into small fragments (Figure B and F), which then co-assembled with CSA into slack nanostructures via electrostatic attraction (Figure C and G). With the prolongation of incubation time, the enhanced electrostatic interaction and the formed hydrogen bonds between DXBTZ-CB[8] and CSA enabled the co-assembled nanoparticles to gradually change from loose to dense, and finally present a spherical shape (Figure D and H). This formation mechanism of DXBTZ-CB[8]/CSA nanoagents explains the transformation of nanorods to nanoparticles, which has been further discussed on Page 13-14 in the revised manuscript.

Co-assembly process of DXBTZ-CB[8] and CSA. Representative transmission electron microscopic images and hydrodynamic diameter distribution for the sample of DXBTZ-CB[8] complexes (10 μ M) before (A and E), and 2 min (B and F), 5 min (C and G) or 30 min (D and H) after co-assembly with CSA (8 μ g mL⁻¹).

The statement on “easy-to-follow method based on supramolecular chemistry” should be weakened as many many reports including those presented in response letter by the authors have worked the supramolecular method and strategy. So the reviewer would like to suggest more emphasis on the performance of optoacoustic contrast agents, which have been well demonstrated in this work.

Our Response: As suggested, we have modified the statement on easy-to-follow method based on supramolecular chemistry and put more emphasis on the optoacoustic performance of constructed nanoagents in the revised manuscript. Thanks for the suggestions.

Overall, the concept delivered in this work is not attractive, but the display on the performance of optoacoustic imaging of assembled nanomaterials (NPs) is of interest. The comments can be a reference for the authors before further consideration for publication.

Our Response: According to the suggestions, we have further revised the manuscript. We sincerely hope that the revised manuscript is now suitable for publication.

Reviewers' Comments:

Reviewer #3:

Remarks to the Author:

No more comments.

Response to Reviewers' Comments

Reviewer #3 (Remarks to the Author):

No more comments.

Our Response: We thank this reviewer for his/her satisfaction with our revised manuscript.